# Large Language Models Are Latent Variable Models: Explaining and Finding Good Demonstrations for In-Context Learning

**Xinyi Wang[1], Wanrong Zhu[1], Michael Saxon[1], Mark Steyvers[2], William Yang Wang[1]**
[1]Department of Computer Science, University of California, Santa Barbara
[2]Department of Cognitive Sciences, University of California, Irvine
{xinyi_wang, wanrongzhu, saxon}@ucsb.edu,
msteyver@uci.edu, william@cs.ucsb.edu

## Abstract

In recent years, pre-trained large language models (LLMs) have demonstrated remarkable efficiency in achieving an inference-time few-shot learning capability known as in-context learning. However, existing literature has highlighted the sensitivity of this capability to the selection of few-shot demonstrations. Current understandings of the underlying mechanisms by which this capability arises from regular language model pretraining objectives remain disconnected from the real-world LLMs. This study aims to examine the in-context learning phenomenon through a Bayesian lens, viewing real-world LLMs as latent variable models. On this premise, we propose an algorithm to select optimal demonstrations from a set of annotated data with a small LM, and then directly generalize the selected demonstrations to larger LMs. We demonstrate significant improvement over baselines, averaged over eight GPT models on eight real-world text classification datasets. We also demonstrate the real-world usefulness of our algorithm on GSM8K, a math word problem dataset. Our empirical findings support our hypothesis that LLMs implicitly infer a latent variable containing task information. [1]

## 1 Introduction

Transformer-based [41] pre-trained large language models (LLMs) have demonstrated significant advancements in a variety of natural language processing (NLP) tasks. As the size of these LLMs increases, they gain "in-context learning" capabilities, whereby the models achieve state-of-the-art (SOTA) or near-SOTA performance by conditioning on a small number of demonstration examples at inference time, without any need for updating model parameters [4]. Below is an example input sequence for semantic analysis with in-context learning:

```
Great movie.  Positive.\n The worst movie ever.  Negative.\n Can't wait to
see the second movie!
```

The first two lines are two demonstrations, and the third line is a test input. We expect an LLM to output the correct label `Positive` as a continuation.

In-context learning has been demonstrated to be an effective technique for a wide range of NLP tasks. However, it is sensitive to the choice, format, and even the order of the demonstrations used [29, 20]. This makes achieving optimal performance with in-context learning a significant challenge, requiring real human effort to adjust the format and selection of demonstration examples. Heuristic solutions, such as selecting demonstrations based on the similarity between the demonstrations and test input

---

[1]Code: https://github.com/WANGXinyiLinda/concept-based-demonstration-selection.

37th Conference on Neural Information Processing Systems (NeurIPS 2023).

[19, 37] have been proposed, but a comprehensive understanding of why certain demonstrations are effective while others are not remains elusive. Additionally, the mechanisms by which LLMs acquire in-context learning capabilities through training on natural text under the standard language model pre-training objective are not fully understood. Recent works on understanding in-context learning provide valuable insights and theoretical results [5, 1, 42, 14, 12], but are limited in scope, focusing on synthetic experiments to validate their hypotheses, while it remains unclear if these results generalize to LLMs pre-trained on real-world natural language data. Xie et al. [50] introduced a prominent result providing a latent topic (concept) variable interpretation for in-context learning. They showed that the in-context learning predictor approaches the Bayes optimal predictor when the number of demonstrations approaches infinity, under the assumption that both the pre-training data distribution and task-specific data distribution are Hidden Markov Models (HMM). However, the assumption that the data generation process is Hidden Markovian makes extrapolation of the result to natural language questionable, and restricts empirical verification to synthetic data with toy models.

We are inspired by this prior work and introduce a more general and natural explanation built on realistic assumptions, which gives rise to a practical demonstration selection algorithm. Our explanation is inspired by the generation process of a topic model, i.e. a simple latent variable model:

$$P(\boldsymbol{w}_{1:T}) = \int_{\Theta} P(\boldsymbol{w}_{1:T}|\boldsymbol{\theta})P(\boldsymbol{\theta})d\boldsymbol{\theta}$$

Where $\boldsymbol{\theta} \in \Theta$ represents a potentially high dimensional topic/concept variable, $\Theta$ is the space of the topic/concept variable, and $\boldsymbol{w}_{1:T}$ refers to the token sequence of a piece of text. Note that the topic model here refers to the modern neural topic models [23, 22]. On the other hand, generative LLMs model text data according to the general probabilistic decomposition:

$$P(\boldsymbol{w}_{1:T}) = \prod_{i=1}^{T} P(\boldsymbol{w}_i|\boldsymbol{w}_{i-1}, ..., \boldsymbol{w}_1)$$

While in practice, LLMs generate new tokens based on all previous tokens, we investigate whether a simplified assumption similar to that of topic models can be made for LLMs:

$$P_M(\boldsymbol{w}_{t+1:T}|\boldsymbol{w}_{1:t}) = \int_{\Theta} P_M(\boldsymbol{w}_{t+1:T}|\boldsymbol{\theta})P_M(\boldsymbol{\theta}|\boldsymbol{w}_{1:t})d\boldsymbol{\theta}$$

In this scenario, the generated tokens are assumed to be conditionally independent of previous tokens, given the latent topic (concept) variable that acts like an approximate sufficient statistic for the posterior information related to the prompt $\boldsymbol{w}_{1:t}$. For in-context learning, this concept variable includes format and task information. By conditioning on an appropriate latent concept variable, LLMs would generate the desired continuation with $P(\boldsymbol{w}_{t+1:T}|\boldsymbol{\theta})$. As LLMs do not explicitly learn a latent variable distribution like LDA-style topic models [3], we can instead utilize this formulation under an Empirical Bayesian formulation inspired by Lester et al. [17] to only approximate the optimal latent variable value for a desired task, using a small LLM (with less than 1B parameters), which is computationally efficient.

We empirically validate our explanation by selecting examples ($\boldsymbol{w}_{1:t}$ in the equations) that are most likely to infer the optimal latent variable value (those with the highest posterior probability $P(\boldsymbol{\theta}|\boldsymbol{w}_{t+1:T})$). We then directly use them as demonstrations for in-context learning with other larger LLMs (up to 175B parameters) and observed a significant performance improvement. The generalization of demonstrations between LLMs is likely a result of similar pre-training data distributions.

While our work is inspired by that of Xie et al. [50], our approach differs significantly in both theoretical analysis and experimental settings. Our main contributions are as follows:

- **We assume a general data generation process** specified by a three-variable causal graph, without constraints on the distribution function or the number of demonstrations.

- **We prove under these realistic assumptions** that the in-context learning predictor can reach the Bayes optimal predictor with a finite number of demonstrations chosen using the latent concept variable.

- **We introduce an efficient, practical demonstration selection algorithm** based on our theoretical results, which can select demonstrations using a small LLM and then directly generalize the demonstrations to other LLMs. The effectiveness of our algorithm is empirically validated using real-world LLMs on both text classification tasks and math word problems.

Our goal is to close the gap between theoretical understandings and real-world LLMs. To the best of our knowledge, our proposed latent variable explanation of in-context learning is the first Bayesian explanation that yields an effective algorithm in real-world scenarios.

## 2 Theoretical Analysis

In in-context learning, the prompt $w_{1:t}$ is composed of several demonstrations and a test input. The generated tokens $w_{t+1:T}$ represent the model's prediction for the test input.

### 2.1 Notations and Problem Setting

Suppose the objective of our task is to predict a discrete target variable $Y \in \mathcal{Y}$, given a token sequence $X \in \mathcal{X}$, where $\mathcal{X}$ is the space of all possible token sequences. $\boldsymbol{\theta} \in \Theta$ is a potentially high dimensional latent variable, where $\Theta$ is the high dimensional space of the variable. Unlike the traditional topic model, $\boldsymbol{\theta}$ is not assumed to be discrete, but continuously distributed over $\Theta$. To define the data generation process, we posit the existence of an underlying causal relation between $X, Y$, and $\boldsymbol{\theta}$. We examine two potential directions of this causal relation, namely $X \to Y \leftarrow \boldsymbol{\theta}$ and $Y \to X \leftarrow \boldsymbol{\theta}$, which can be represented mathematically as the following structural equations:

$$Y = f(X, \boldsymbol{\theta}, \boldsymbol{\epsilon}) \qquad\qquad X = g(Y, \boldsymbol{\theta}, \boldsymbol{\epsilon})$$

Here $\boldsymbol{\epsilon} \in \mathcal{E}$ is an independent noise variable, $f : \mathcal{X} \times \Theta \times \mathcal{E} \to \mathcal{Y}$ and $g : \mathcal{Y} \times \Theta \times \mathcal{E} \to \mathcal{X}$ are two deterministic functions. Furthermore, we denote the joint data distribution by $X, Y, \boldsymbol{\theta} \sim P$, and assume that $Y$ is sampled from a uniform distribution over $\mathcal{Y}$. The distinction between these two directions is crucial, as it allows us to utilize the direction in which the child variable ($Y$ or $X$) is independent of the other variables, given its parents.

We hypothesize that the causal direction depends on the nature of the task. For instance, in the task of predicting the sentiment ($Y$) of a movie review ($X$), it is reasonable to assume that the opinion about the movie is formed before writing the review, thus making $Y$ the cause of $X$, along with the task concept of "writing a passage to express one's opinion about the movie" ($\boldsymbol{\theta}$). Conversely, for the task of classifying whether a product review ($X$) is helpful to other customers ($Y$), it is the quality of the review ($X$) that cause other customers to upvote it ($Y$), along with the task concept of "rating the helpfulness of this review" ($\boldsymbol{\theta}$). *In the rest of the paper, we will focus on the $X \to Y \leftarrow \boldsymbol{\theta}$ direction and leave a detailed discussion of the other direction in the Appendix.*

Suppose we are interested in a task (e.g. semantic analysis) denoted by $d \in \mathcal{T}$, where $\mathcal{T}$ is the space of all possible tasks. We assume there is an injective function between $\mathcal{T}$ and $\Theta$. i.e. for each task $d$, there is a concept variable $\theta^d$, such that each data $(X^d, Y^d)$ sampled from task $d$ is generated by:

$$Y^d = f(X^d, \theta^d, \boldsymbol{\epsilon})$$

To perform in-context learning with an LLM (generically denoted by model label $M$), we condition on a fixed set of $k$ demonstration examples $(X_1^d, Y_1^d), (X_2^d, Y_2^d), ..., (X_k^d, Y_k^d)$ sampled from task $d$.

Following previous works [24, 26], as we are not using any instruction fine-tuned models, we do not include a task description in the prompt, with the aim of focusing on the examination of the demonstrations. To naturally project $\mathcal{Y}$ into the token space $\mathcal{X}$, we define injective mappings $\tau^d : \mathcal{Y} \to \mathcal{X}$, which are typically defined by human understanding of the task $d$. e.g. for sentiment analysis, $\tau^d$ map positive class to the token "positive" and negative class to the token "negative". Additionally, a delimiter token $\boldsymbol{w}^d$ is defined, typically an empty space or a new line token, to separate the demonstrations when concatenated. We denote the LLM output probability of $X, Y$, and $\boldsymbol{\theta}$, with the aforementioned preprocessing applied, by $P_M^d$:

$$P_M(\tau^d(Y) | X_1^d, \tau^d(Y_1^d), \boldsymbol{w}^d, ..., X_k^d, \tau^d(Y_k^d), \boldsymbol{w}^d, X) = P_M^d(Y | X_1^d, Y_1^d, ..., X_k^d, Y_k^d, X)$$

### 2.2 Problem Analysis and Theoretical Results

Suppose a set of observed data sampled from task $d$, denoted as $\mathcal{D}^d$, is available, allowing for the selection of the $k$ most suitable demonstrations from it. For any incoming test example $X$, we have:

$$P_M^d(Y | X_1^d, Y_1^d, ..., X_k^d, Y_k^d, X) = \int_\Theta P_M^d(Y | \boldsymbol{\theta}, X) P_M^d(\boldsymbol{\theta} | X_1^d, Y_1^d, ..., X_k^d, Y_k^d, X) d\boldsymbol{\theta} \qquad (1)$$

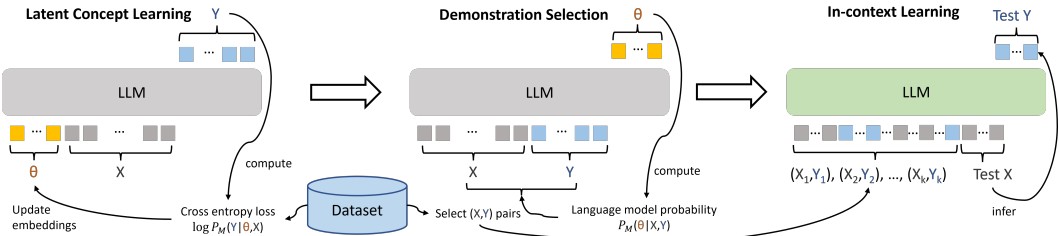

Figure 1: An overview of our proposed two-phased algorithm. Demonstration selection and latent concept learning share the same LLM as demonstration selection needs to reuse the learned concept tokens, while at the in-context learning time, any other generative LLMs can be used. Here we only illustrate the $X \to Y \leftarrow \boldsymbol{\theta}$ direction. The $Y \to X \leftarrow \boldsymbol{\theta}$ direction can be illustrated similarly by exchanging $X$ and $Y$ in the above figure.

Here, we assume the sampling of the test example is independent of the sampling of the demonstrations, so $Y$ is independent of the demonstrations given $\boldsymbol{\theta}$ and $X$. We also assume that the pre-trained data distribution $P_M^d$ is a suitable approximation of the assumed data distribution $P$:

**Assumption 2.1.** Assume that $P_M(X) = P(X)$, and $P_M^d(Y|\boldsymbol{\theta}, X) \propto P(Y|\boldsymbol{\theta}, X)$ for $X \to Y \leftarrow \boldsymbol{\theta}$.

Note that the assumption that a large language model captures the true distribution of language is fairly common in the literature studying LLMs [50, 34, 47]. With this assumption, we establish:

**Proposition 2.2.** *If task $d$ follows the $X \to Y \leftarrow \boldsymbol{\theta}$ direction, then $\arg \max_{y \in \mathcal{Y}} P_M^d(Y = y|\theta^d, X)$ is the Bayes optimal classifier.*

In this case, only when $P_M^d(\boldsymbol{\theta}|X_1^d, Y_1^d, ..., X_k^d, Y_k^d, X)$ completely concentrate on $\theta^d$, can the in-context learning classifier become the Bayes optimal classifier [11]:

**Theorem 2.3.** *If task $d$ follows the $X \to Y \leftarrow \boldsymbol{\theta}$ direction, then the in-context learning classifier*

$$\arg \max_{y \in \mathcal{Y}} P_M^d(Y = y|X_1^d, Y_1^d, ..., X_k^d, Y_k^d, X)$$

*always has a higher or equal probability of misclassification to the Bayes optimal classifier $\arg \max_{y \in \mathcal{Y}} P_M^d(Y = y|\theta^d, X)$. Equality only holds when*

$$\forall x \in \mathcal{X}, \ P_M^d(\theta^d|X_1^d, Y_1^d, ..., X_k^d, Y_k^d, X = x) = 1.$$

A similar argument can be made for the $Y \to X \leftarrow \boldsymbol{\theta}$ direction. [2] Here, Equation (1) would become:

$$P_M^d(X|Y_1^d, X_1^d, ..., Y_k^d, X_k^d, Y) = \int_{\Theta} P_M^d(X|\boldsymbol{\theta}, Y) P_M^d(\boldsymbol{\theta}|Y_1^d, X_1^d, ..., Y_k^d, X_k^d, Y) d\boldsymbol{\theta} \quad (2)$$

Note that the left-hand side of Equation (1) and Equation (2) are similar to the direct and channel method introduced by Min et al. [24]. However, our analysis differs from theirs in that we do not treat $(Y \to X \leftarrow \boldsymbol{\theta})$ as the universally superior channel direction for modeling in-context learning, rather arguing that depending on the end task, the causal direction $(X \to Y \leftarrow \boldsymbol{\theta})$ is sometimes better. This view is supported by our empirical results in Appendix B.

## 3 Method

Here we demonstrate how the proposed theory can be practically applied to select optimal demonstration examples. Since latent variable $\boldsymbol{\theta}$ encodes both the task and format information, the whole distribution over $\Theta$ is too complex to model. Unlike traditional topic models, we will only focus on estimating an optimal value $\theta^d$ corresponding to task $d$.

First, we perform *latent concept learning*, wherein the task latent $\theta^d$ is learned as a set of new token embeddings using prompt tuning over the full demonstration candidate set. With this optimal task latent, we then perform *demonstration selection*, where a smaller set of demonstrations is chosen to maximize the likelihood of postfixing the latent concept tokens. We only need to use a small LLM to do the above steps to obtain an optimal set of demonstrations that can be directly transferred to other LLMs. Figure 1 is an overall illustration of our proposed method.

---

[2]The detailed argument of the $Y \to X \leftarrow \boldsymbol{\theta}$ direction can be found in Appendix A.2.

---

**Algorithm 1** Latent concept learning

---

**Input:** Dataset $\mathcal{D} = \{(x_i, y_i, d_i)\}_i$ associated with a set of tasks $\mathcal{S}$, LLM $M$, number of concept tokens per task $c$, learning rate $\alpha$, and number of training steps $N$.
**Output:** LLM $M'$ with fine-tuned concept tokens.
Add $c|\mathcal{S}|$ new tokens to the vocabulary. i.e. The concept tokens $\hat{\theta}^d$ for each task in $\mathcal{S}$. Randomly initialize their embeddings $E_{new}$. Freeze all parameters in $M$ except $E_{new}$;
**for** step $= 1$ **to** $N$ **do**
    Sample a random batch $B$ in $\mathcal{D}$ and initialize gradient $g \leftarrow 0$;
    **for** each data point $(x, y, d)$ in $B$ **do**
        $g = g + \frac{\partial \ell(X, Y; \hat{\theta}^d)}{\partial E_{new}}$;
    **end for**
    $E_{new} = E_{new} - \alpha g$;
**end for**

---

### 3.1 Latent Concept Learning

We want to first find the optimal value of the latent concept variable $\theta^d$ corresponding to a task $d \in \mathcal{T}$. As $\arg\max_{y \in \mathcal{Y}} P_M^d(Y = y|\theta^d, X)$ is the Bayes optimal classifier according to Proposition 2.2, $\theta^d$ should be able to minimize $-\mathbb{E}_{X,Y,d}[\log P_M^d(Y|\theta^d, X)]$ for the $X \rightarrow Y \leftarrow \boldsymbol{\theta}$ direction. In practice, we try to align $\theta^d$ to the token embedding space by adding new tokens to the vocabulary. After this alignment, we hope to be able to use the learned new tokens of $\theta^d$ as regular tokens.

More specifically, building upon the methodology proposed by Lester et al. [17], for each specific task $d$, $c$ new concept tokens (denoted as $\hat{\theta}^d$) are added to the original vocabulary of LLM $M$ to represent the corresponding task concept $\theta^d$. Subsequently, the embedding of these new tokens $E_{new}(\hat{\theta}^d)$ is fine-tuned while freezing the remaining parameters of LLM $M$. The variable $c$ is treated as a hyperparameter. In practice, in order to condition on $\theta^d$, the corresponding $c$ concept tokens are appended to the input $X$ (or $Y$) as shown in the example provided below, where $c = 2$:

```
<sentiment_token_1><sentiment_token_2> Can't wait to see the second movie!
```

By giving the above input tokens, we ask the LLM to predict the correct label `Positive` for us. Note that `<sentiment_token_1>` here is just a label assigned to the newly added concept token. It can be anything as long as it does not overlap with the original vocabulary of LLM.

The fine-tuning objective would then be minimizing $\mathcal{L}(\hat{\theta}^d) = \mathbb{E}_{X,Y}[\ell(X, Y; \hat{\theta}^d)]$, where

$$\ell(X, Y; \hat{\theta}^d) = \begin{cases} -\log P_M^d(Y|\hat{\theta}^d, X) & \text{if } X \rightarrow Y \leftarrow \boldsymbol{\theta} \\ -\log P_M^d(X|\hat{\theta}^d, Y) & \text{if } Y \rightarrow X \leftarrow \boldsymbol{\theta}. \end{cases}$$

Theoretically, if we can minimize the above loss function, a Bayes optimal classifier can be obtained, and the concept tokens would be a reasonable delegate of the real latent concept variable:

**Proposition 3.1.** *When $\mathcal{L}(\hat{\theta}^d)$ is minimized, $P_M^d(Y|\hat{\theta}^d, X) = P(Y|\theta^d, X)$ for $X \rightarrow Y \leftarrow \boldsymbol{\theta}$. If the LLM $M$ is invertible, then $\hat{\theta}^d = \theta^d$.*[3]

We denote the LLM $M$ with fine-tuned concept tokens by $M'$. Since we add the concept tokens into the regular token vocabulary, the raw LLM output probability $P_{M'}(\hat{\theta}^d|\boldsymbol{w}_{1:t})$ ($\boldsymbol{w}_{1:t}$ denote a given prompt) would be in the token sequence space $\mathcal{X}$ instead of the concept space $\Theta$. Since learning all possible $\theta^d \in \Theta$ is infeasible, we propose to approximate the concept space $\Theta$ by sampling a diverse subset of tasks $\mathcal{S} \subseteq \mathcal{T}$. Then the estimated conditional probability of $\theta^d$ would be:

$$\hat{P}_{M'}^d(\hat{\theta}^d|\boldsymbol{w}_{1:t}) = \frac{P_{M'}^d(\hat{\theta}^d|\boldsymbol{w}_{1:t})}{\sum_{t \in \mathcal{S}} P_{M'}^t(\hat{\theta}^t|\boldsymbol{w}_{1:t})}$$

To obtain the concept tokens for all tasks in $\mathcal{S}$, we fine-tune all tasks together with the loss $\sum_{d \in \mathcal{S}} \mathcal{L}(\theta^d)$. We summarize the proposed algorithm in Algorithm 1.

---

[3]More discussion can be found in Appendix A.3.

**Algorithm 2** Demonstration selection

---

**Input:** dataset $\mathcal{D}^d$ for a task $d$. LLM with fine-tuned concept tokens $M'$. The number of demonstrations $k$.
**Output:** A set of selected demonstrations.
**for** each $(X^d, Y^d)$ in $\mathcal{D}^d$ **do**
    Compute $\hat{P}_M^d(\hat{\theta}^d | X^d, Y^d)$;
**end for**
Select top $k$ examples with the largest $\hat{P}_M^d(\hat{\theta}^d | X^d, Y^d)$, denoted as $(X_1^d, Y_1^d), ..., (X_k^d, Y_k^d)$;

---

Note that the embedding matrix of a generative LLM is shared on both the input and output sides. So while we only see the concept tokens on the input side at the training time, they can be viewed as regular word tokens that can be generated on the output side.

## 3.2 Demonstration Selection

According to Theorem 2.3, for a task $d$, to make the in-context learning classifier closer to the Bayes optimal classifier, we need to select demonstrations $(X_1^d, Y_1^d), ..., (X_k^d, Y_k^d)$ that maximize $P_M^d(\theta^d | X_1^d, Y_1^d, ..., X_k^d, Y_k^d, X)$ for all $X \in \mathcal{X}$. Then our goal then becomes selecting demonstrations that can best infer the task concept for all test inputs on average:

$$\underset{X_1^d, Y_1^d, ..., X_k^d, Y_k^d}{\arg\max} \mathbb{E}_X[P_M^d(\theta^d | X_1^d, Y_1^d, ..., X_k^d, Y_k^d, X)]$$

As test examples are sampled independent of the demonstrations, and $P_M(X) = P(X)$ according to Assumption 2.1, we have

$$\mathbb{E}_X[P_M^d(\theta^d | X_1^d, Y_1^d, ..., X_k^d, Y_k^d, X)] = P_M^d(\theta^d | X_1^d, Y_1^d, ..., X_k^d, Y_k^d)$$

If we assume each demonstration is also sampled independently, we have:

$$P_M^d(\theta^d | X_1^d, Y_1^d, ..., X_k^d, Y_k^d) = \frac{\prod_{i=1}^k P_M^d(\theta^d | X_i^d, Y_i^d)}{P_M^d(\theta^d)^{k-1}}$$

Assuming that $\theta$ has a uniform prior, then our goal becomes finding the top $k$ demonstrations that maximize $\hat{P}_{M'}^d(\hat{\theta}^d | X_i^d, Y_i^d)$. Note that the independence between demonstrations is a simplified assumption to reduce the combinatory search space of $(X_1^d, Y_1^d), ..., (X_k^d, Y_k^d)$. In practice, selected demonstrations are likely correlated as some demonstrations may work well together but not necessarily work well by themselves. However, it would be too expensive to search the $O(|\mathcal{D}^d|^k)$ combinations over the candidate set $\mathcal{D}^d$. In practice, this simplification works reasonably well. We leave this combinatory search problem to future research.

Also, as we are using an LLM to approximate the data distribution, the order of the demonstrations might matter. We will show in the Experiment section that the order does not matter, so no reordering of the selected demonstrations is needed. The full selection algorithm is shown in Algorithm 2.

# 4 Experiments

**Datasets.** We conduct experiments on eight datasets from five different types of NLP classification tasks: sentiment analysis, linguistic analysis, topic classification, emotion classification, and hate speech detection. For sentiment analysis, we choose the Stanford Sentiment Treebank (SST2) dataset [35] from the GLUE benchmark [43] and the financial phrase bank (FPB) dataset [21]. SST2 is constructed based on movie reviews labeled "positive" or "negative", and FPB is based on financial news labeled "positive", "negative", or "neutral". For linguistic analysis, we choose the Corpus of Linguistic Acceptability (COLA) dataset [46] from the GLUE benchmark, based on sentences collected from linguistic books, labeled with "acceptable" or "unacceptable". For topic classification, we choose the DBpedia ontology classification dataset [52], based on DBpedia 2014 [16], labeled with 14 different ontology classes. For emotion classification, we choose the dataset from Chatterjee et al. [6] and Saravia et al. [33], both of which are collected from Twitter. Chatterjee et al. [6] (EmoC)

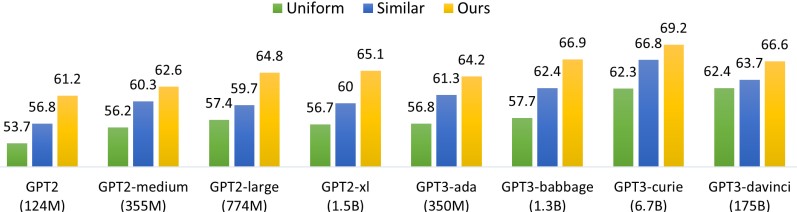

Figure 2: Accuracy of 4-shot in-context learning using demonstrations selected by our method and other baselines, averaged over eight datasets. Our demonstrations are selected using GPT2-large, and the same set of demonstrations is then applied to all other LLMs.

predict emotion given a three-turn contextual dialogue, while Saravia et al. [33] predict emotion given a Twitter message with clear emotion. For hate speech detection, we choose the online hate speech detection dataset (ETHOS) [27], collected from online social media platforms. Here we detect two types of hate speech: sexual orientation (ETHOS-SO) and religion (ETHOS-R). While in Section 2, we assume that all tasks share the same label space $\mathcal{Y}$, here we relax such assumption and allow a different number of labels for different tasks. We use minimal formatting to process each example. A detailed description of the datasets and our data processing procedure can be found in Appendix B.

**Experiment settings.** To determine the causal direction for each task, we select the direction that can give higher accuracy when using random demonstrations[4]. We adopt the $Y \rightarrow X \leftarrow \boldsymbol{\theta}$ direction for sentiment analysis, topic classification, and emotion classification tasks, which is consistent with the intuition that people usually have some sentiment, topic, or emotion in mind before writing a piece of text. We adopt the $X \rightarrow Y \leftarrow \boldsymbol{\theta}$ direction for the linguistic analysis and hate speech detection type of tasks. While this is less intuitive, we can understand this as linguistic error and hate speech detection are more of a post hoc task in contrast to the previous tasks.

Without specification, we use $k = 4$ number of demonstrations and $c = 10$ number of concept tokens per dataset for our experiments, as the context length of GPT2 is 1024, and a larger number of demonstrations may not be able to completely fit into it. We use GPT2-large to learn the concept tokens and then compute the probability of each candidate demonstration example. We select our demonstrations from a randomly selected 100 example subset of the train set as the candidate set $\mathcal{D}^d$. We use the same set of demonstrations selected by GPT2-large for all other LLMs. We test the performance of the selected demonstrations using at most 1000 examples randomly sampled from the test set. Each experiment is repeated for five runs with different random seeds (the randomness comes from the sampling of the candidate set and the sampling of the test set). We adopt a large portion of the code from Min et al. [25], which is based on Huggingface [49].

**Baselines.** We consider the following baselines:

- **Uniform**: We uniformly select $k$ demonstrations from $\mathcal{D}$ for each test example.
- **Similar**: According to Liu et al. [19], demonstrations that are semantically similar to the test example would hare more performant. Following their method, we use a pre-trained sentence Transformer [31] to calculate the cosine similarity between the demonstrations and test examples. We choose the top $k$ similar demonstrations from $\mathcal{D}$ for each test example.

**Main results.**[5] Figure 2 shows our main results averaged over all eight datasets, using the first-generation GPT2s and GPT3s, without any instruction fine-tuning [28] or Reinforcement Learning from Human Feedback (RLHF) [36]. Our method significantly outperforms baselines on eight different LLMs, with 12.5% relative improvement to the uniform selection baseline on average, which shows the effectiveness of our method. The demonstrations selected by our method are exclusively based on GPT2-large, while the same set of demonstrations can be generalized to all other GPTs.

**Results with non-GPT models.** In Figure 3a, we test the demonstrations selected by our method using GPT2-large on more LLMs (GPT3 [4], GPT3-instruct [28, 36], GPT-J [44], OPT [51], and LLaMA [38]) with similar sizes (6-7B), and show that the selected demonstrations improve in-context learning performance of all of them. The fact that GPT3-curie obtains the largest performance improvement is likely because similar pre-training data distributions help the generalization of the

---

[4]Detailed results see Figure 6 in Appendix B.

[5]The complete results with standard deviations in this section can be found in Appendix B.

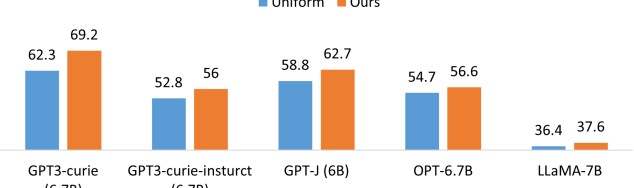
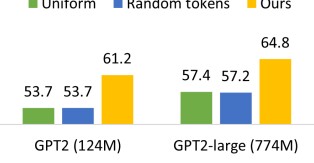

(a) Proposed method v.s. randomly selected demonstrations

(b) Proposed method v.s. using randomly selected tokens

Figure 3: In-context learning accuracy averaged over all eight datasets.

|  | Uniform | Similar | Ours w/ Llama 2 (7B) | Ours w/ GPT2-XL (1.5B) |
|---|---|---|---|---|
| Prompt tuning | - | - | 3.7 | 1.3 |
| Llama 2 (7B) | 11.4 | 13.1 | 19.3 | 15.9 |
| Llama 2 (13B) | 17.0 | 18.3 | 21.6 | 20.5 |
| Llama 2 (70B) | 50.2 | 53.5 | 54.3 | 52.9 |
| ChatGPT (gpt-3.5-turbo) | 76.5 | 78.1 | 81.2 | 80.4 |

Table 1: Prompt tuning and 4-shot in-context learning accuracy on a subset of GSM8K test set. Our demonstrations are selected with either 7B Llama 2 or GPT2-XL

selected demonstrations. Different-size GPT2 models share the same pre-training corpus [30], while GPT3s are pre-trained on a dataset expanded from the GPT2 pre-training corpus [4]. Thus the pre-training distribution of GPT3-curie and GPT2-large can be assumed to be similar.

**Results on GSM8K.** Since our primary goal is to connect the theory with real-world models and datasets, we did not try to include harder tasks in the main results in Figure 2. In practice, our proposed method is most effective with hard tasks that even parameter-efficient fine-tuning with smaller models cannot outperform in-context learning with the same or larger models. To showcase the usefulness of our proposed algorithm, We added a new dataset, GSM8K [9], which is a math word problem-solving dataset with chain-of-thoughts solutions. Table 1 shows the test accuracy of the final numerical answer with greedy generation. We randomly select a test set of 200 examples instead of using the full test set for computation efficiency. [6]

As shown in the first row of Table 1, prompt tuning with ten new tokens can only obtain less than 4% accuracy on the GSM8K test set. The last four rows show the in-context learning results with different size Llama 2 models [39] and ChatGPT. Our proposed demonstration selection method (last two columns) significantly outperformed the Uniform and Similar baseline. We also find that the demonstrations selected with a larger model (7B) are more effective than those selected with a smaller model (1.5B). The results show that our demonstration selection method is a good choice under a low data setting, with a small computing budget and minimal inference latency. Our proposed method can also potentially be combined with other prompting techniques [8] to boost performance further.

**Learned tokens v.s. Random tokens.** To confirm the critical role of the latent concept variable in the proposed demonstration selection algorithm, we compare the performance of using the learned concept tokens versus using randomly selected tokens from the original vocabulary in Figure 3b. The demonstrations selected by random tokens only obtain the same performance as randomly selected demonstrations, showing that the performance gain of our method comes from the learned concept tokens containing the task and format information, not other elements of our algorithm.

$k$ **ablation study.** While we use $k = 4$ demonstrations for all experiments, we also test the effectiveness of our method using different $k$. As shown in Figure 4a, our method significantly outperforms the random selection baseline with $k = 2, 4, 8$, and 16. To fit in large $k$s, we use GPT3-ada with a longer context length (2048). Note that for real-world tasks, it is in general not true that more demonstrations guarantee higher performance [7]. We can see that the uniform baseline performance increases from $k = 2$ to $k = 8$, then drops a little at $k = 16$. Our method improves the uniform baseline by around 5% absolute for all $k$s, while $k = 4$ improves the most (6.6%). Our method appears to have a diminishing effect when $k$ becomes larger, which is likely because the effect of more demonstrations overwhelms the effect of demonstration choices.

---

[6]Note that we did not use a calculator to insert the correct result of each generated math equation during generation for time efficiency, which resulted in slightly lower scores.

$c$ **ablation study.** While we use $c = 10$ number of concept tokens for all experiments, we also investigate the effect of different $c$ on our method. When $c$ is small ($c = 5$), the concept tokens cannot effectively capture the task and format information, thus cannot im-

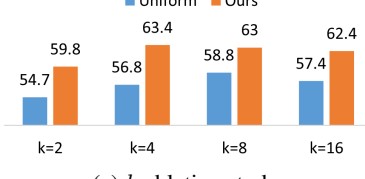 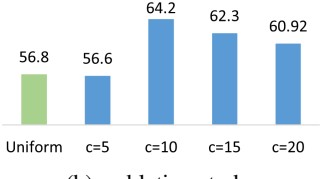

(a) $k$ ablation study.  (b) $c$ ablation study.

Figure 4: In-context learning accuracy of our method versus random selection baseline averaged over all eight datasets with GPT3-ada.

prove the performance. When $c$ increases from 10 to 20, we observe a drop in the performance. It is likely because the selectivity of the concept tokens decreases when $c$ increases. The longer the concept token sequence is, the more likely it will contain meaningless tokens that do not contribute to demonstration selection.

**Effect of demonstrations' order.** We find that the demonstrations selected by our method are insensitive to their order in most cases.[7] An exception is the EmoC dataset, where our method has a high variance. On the contrary, Lu et al. [20] found that the order of the demonstration matters, and a good ordering cannot be transferred between different LLMs. We suspect that the ordering only matters when the demonstration selection method is not robust. Since Lu et al. [20] randomly selects one set of demonstrations for the whole test set, the variance in performance is high with different demonstrations, thus ordering matters. And since such ordering is not transferable while our selected demonstrations are highly transferable, we suspect the core task information is stored in the content of the demonstrations, while the ordering mainly captures model-specific artifacts.

**Qualitative analysis.** In Figure 5, we provide a t-SNE [40] projection of the learned concept token embeddings. The tokens corresponding to semantically similar tasks are close together. Note that this result only aims to provide a straightforward illustration of concept tokens. The effect of concept tokens should be understood by the previous quantitative results.[8]

We also list the top 4 selected demonstrations in Table 14 in Appendix B. Compared to the examples with lower scores, the selected examples for GSM8K have more deductive reasoning (i.e. with the connecting words 'so', 'then', 'thus', etc.), instead of listing parallel conditions. For SST2, the selected examples are longer and more complex, sometimes including a 'but'. This can be understood as these

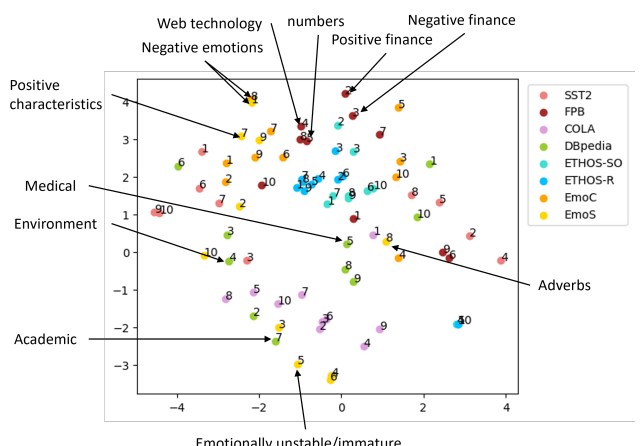

Figure 5: t-SNE plot of the learned concept tokens for each task. Concept tokens that can be explained by similar tokens are summarized in the graph.

harder examples can represent the task more comprehensively. This conclusion also aligns with the findings in [13] that hard examples in the pre-training data contribute to in-context learning the most. The label distribution of the selected demonstrations is usually balanced in class, which reduces the possible biases introduced by the demonstrations.

# 5 Related Work

Heuristic solutions, such as selecting demonstrations based on the similarity between the demonstrations and test input [19, 37, 32] have been proposed. [20] propose to reorder the demonstration based on the entropy of the predicted labels. In this paper, we use the similarity-based selection method

---

[7]Detailed results see Figure 9 in Appendix B.

[8]The list of similar tokens for these concept tokens can be found in Table 13 in Appendix B.

as a baseline while do not include the label entropy-based reordering method as we show that the ordering of the demonstrations does not matter for our method.

Previous research on the phenomenon of in-context learning in Transformers has identified a number of pre-training data distributions that can lead to the emergence of this capability, including a Hidden Markov Model distribution [50] and a skewed Zipfian distribution with high burstiness [5]. Other studies have sought to understand the underlying mechanisms of in-context learning by making connections with gradient descent [42, 10, 1], formalizing it as an algorithm learning problem [18], or proposing a latent variable theory similar as ours [14, 12, 50]. While providing valuable insights on how in-context learning works, these works are limited to synthetic datasets and toy Transformers, while it remains unclear if these results generalize to LLMs pre-trained on real-world text data and whether these results can help in-context learning performance. In contrast, we propose a Bayesian explanation of in-context learning that can be verified with real-world LLMs on various NLP datasets. Dai et al. [10] provide a practical algorithm based on the understanding that the Transformer has a dual form of gradient descent. However, their empirical results are smaller in scale, with six datasets and only one model (350M), and has less significant improvements (5.4% relative to baseline).

There are also works trying to understand in-context learning from an empirical perspective [2, 24]. Min et al. [26] found demonstrations' ground truth labels do not matter for in-context learning, which we find is not entirely accurate in Appendix B. On the other hand, chain-of-thoughts prompting [48, 53, 45] find that providing step-by-step explanations improves in-context learning performance.

## 6 Conclusion

In this work, we endeavor to comprehend large language models (LLMs) through a Bayesian lens and posit them as implicit topic models that infer a latent conceptual variable from prompts. Motivated by this understanding, we propose a two-step algorithm that first extracts latent conceptual tokens from a small LLM and then selects demonstrations that have the greatest probability of predicting the corresponding conceptual tokens. The selected demonstrations can then be directly generalized to other LLMs. The efficacy of our algorithm across various text classification datasets and GPT models validates our explanation of in-context learning.

## Acknowledgements

This work was supported by the National Science Foundation award #2048122. The views expressed are those of the author and do not reflect the official policy or position of the US government. We thank Google for its generous gift to the University of California.

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

# A  Proofs

## A.1  Direct direction

**Assumption A.1.** (Assumption 2.1) Assume that $P_M(X) = P(X)$, and $P_M^d(Y|\boldsymbol{\theta}, X) \propto P(Y|\boldsymbol{\theta}, X)$ for $X \to Y \leftarrow \boldsymbol{\theta}$.

**Proposition A.2.** *(Proposition 2.2) If task $d$ follows the $X \to Y \leftarrow \boldsymbol{\theta}$ direction,* $\arg\max_{y \in \mathcal{Y}} P_M^d(Y = y|\theta^d, X)$ *is the Bayes optimal classifier.*

*Proof.* Since the data generation of the task $d$ can be written as $Y = f(X, \theta^d, \boldsymbol{\epsilon})$, we have

$$P^d(Y|X) = P(Y|\theta^d, X).$$

And by Assumption A.1, we have

$$\arg\max_{y \in \mathcal{Y}} P_M^d(Y = y|\theta^d, X) = \arg\max_{y \in \mathcal{Y}} P(Y = y|\theta^d, X).$$

Thus $\arg\max_{y \in \mathcal{Y}} P_M^d(Y = y|\theta^d, X)$ is the Bayes optimal classifier. $\qquad\square$

**Theorem A.3.** *(Theorem 2.3) If task $d$ follows the $X \to Y \leftarrow \boldsymbol{\theta}$ direction, then the in-context learning classifier*

$$\arg\max_{y \in \mathcal{Y}} P_M^d(Y = y|X_1^d, Y_1^d, ..., X_k^d, Y_k^d, X)$$

*always has a higher or equal probability of misclassification to the Bayes optimal classifier* $\arg\max_{y \in \mathcal{Y}} P_M^d(Y = y|\theta^d, X)$. *Equality only takes when*

$$\forall x \in \mathcal{X}, \ P_M^d(\theta^d|X_1^d, Y_1^d, ..., X_k^d, Y_k^d, X = x) = 1.$$

*Proof.* Recall that in Equation (1), we have

$$P_M^d(Y|X_1^d, Y_1^d, ..., X_k^d, Y_k^d, X) = \int_\Theta P_M^d(Y|\boldsymbol{\theta}, X) P_M^d(\boldsymbol{\theta}|X_1^d, Y_1^d, ..., X_k^d, Y_k^d, X) d\boldsymbol{\theta}.$$

By Proposition A.2, $\arg\max_{y \in \mathcal{Y}} P_M^d(Y = y|\theta^d, X)$ is the Bayes optimal classifier. Let $C_{\boldsymbol{\theta}}(X) = \arg\max_{y \in \mathcal{Y}} P_M^d(Y = y|\boldsymbol{\theta}, X)$, then the risk is defined as the probability of misclassification

$$R(C_{\boldsymbol{\theta}}) = P(C_{\boldsymbol{\theta}}(X) \neq Y) = \mathbb{E}_{XY}[\mathbb{1}_{C_{\boldsymbol{\theta}}(X) \neq Y}].$$

Denote the in-context learning classifier $\arg\max_{y \in \mathcal{Y}} P_M^d(Y = y|X_1^d, Y_1^d, ..., X_k^d, Y_k^d, X)$ by $C_k(X)$. We then have

$$R(C_k) = \mathbb{E}_{XY}[\mathbb{1}_{C_k(X) \neq Y}] = \mathbb{E}_X[\sum_{y \in \mathcal{Y}} (1 - P_M^d(Y = y|\theta^d, X))\mathbb{1}_{C_k(X) = y}].$$

Such risk is minimized if and only if $C_k(X) = C_{\theta^d}(X)$, which only holds when $P_M^d(\theta^d|X_1^d, Y_1^d, ..., X_k^d, Y_k^d, X = x) = 1$ for all $x \in \mathcal{X}$. $\qquad\square$

## A.2  Channel direction

**Assumption A.4.** Assume that $P_M(X) = P(X)$, and $P_M^d(X|\boldsymbol{\theta}, Y) \propto P(X|\boldsymbol{\theta}, Y)$ for the $Y \to X \leftarrow \boldsymbol{\theta}$ direction.

**Proposition A.5.** *If task $d$ follows the $Y \to X \leftarrow \boldsymbol{\theta}$ causal direction,* $\arg\max_{y \in \mathcal{Y}} P_M^d(X|\theta^d, Y = y)$ *is the Bayes optimal classifier when the label assignment is balanced.*

*Proof.* Since the data generation of the task $d$ can be written as $X = g(Y, \theta^d, \boldsymbol{\epsilon})$, we have

$$P^d(X|Y) = P(X|\theta^d, Y)$$

When the label is balanced, i.e. $P^d(Y) = \frac{1}{|\mathcal{Y}|}$, we have

$$P^d(Y|X) = \frac{P^d(X|Y)P^d(Y)}{P(X)} \propto P^d(X|Y)$$

And by Assumption A.4, we have

$$\arg\max_{y \in \mathcal{Y}} P_M^d(X|\theta^d, Y = y) = \arg\max_{y \in \mathcal{Y}} P(X|\theta^d, Y = y).$$

Thus $\arg\max_{y \in \mathcal{Y}} P_M^d(X|\theta^d, Y = y) = \arg\max_{y \in \mathcal{Y}} P^d(Y = y|X)$ is the Bayes optimal classifier.
$\square$

**Theorem A.6.** *If task $d$ follows the $Y \to X \leftarrow \boldsymbol{\theta}$ direction, then the in-context learning classifier*

$$\arg\max_{y \in \mathcal{Y}} P_M^d(X|Y_1^d, X_1^d, ..., Y_k^d, X_k^d, Y = y)$$

*always has a higher or equal probability of misclassification to the Bayes optimal classifier* $\arg\max_{y \in \mathcal{Y}} P_M^d(X|\theta^d, Y = y)$. *Equality only takes when*

$$\forall y \in \mathcal{Y}, \ P_M^d(\theta^d|Y_1^d, X_1^d, ..., Y_k^d, X_k^d, Y = y) = 1.$$

*Proof.* This theorem can be proved similarly as Theorem A.3. Recall that in Equation (2), we have

$$P_M^d(X|Y_1^d, X_1^d, ..., Y_k^d, X_k^d, Y) = \int_\Theta P_M^d(X|\boldsymbol{\theta}, Y) P_M^d(\boldsymbol{\theta}|Y_1^d, X_1^d, ..., Y_k^d, X_k^d, Y) d\boldsymbol{\theta}.$$

By Proposition A.5, $\arg\max_{y \in \mathcal{Y}} P_M^d(X|\theta^d, Y = y)$ is the Bayes optimal classifier. Let $C_{\boldsymbol{\theta}}(X) = \arg\max_{y \in \mathcal{Y}} P_M^d(X|\boldsymbol{\theta}, Y = y)$, then the risk is defined as the probability of misclassification

$$R(C_{\boldsymbol{\theta}}) = P(C_{\boldsymbol{\theta}}(X) \neq Y) = \mathbb{E}_{XY}[\mathbb{1}_{C_{\boldsymbol{\theta}}(X) \neq Y}].$$

Denote the in-context learning classifier $\arg\max_{y \in \mathcal{Y}} P_M^d(X|Y_1^d, X_1^d, ..., Y_k^d, X_k^d, Y = y)$ by $C_k(X)$. We then have

$$R(C_k) = \mathbb{E}_{XY}[\mathbb{1}_{C_k(X) \neq Y}] = \mathbb{E}_X[\sum_{y \in \mathcal{Y}} (1 - P_M^d(X|\theta^d, Y = y))\mathbb{1}_{C_k(X) = y}].$$

Such risk is minimized if and only if $C_k(X) = C_{\theta^d}(X)$, which only holds when $P_M^d(\theta^d|Y_1^d, X_1^d, ..., Y_k^d, X_k^d, Y = y) = 1$ for all $y \in \mathcal{Y}$.
$\square$

## A.3 Method

**Proposition A.7.** *(Proposition 3.1) When $\mathcal{L}(\hat{\theta}^d)$ is minimized, $P_M^d(Y|\hat{\theta}^d, X) = P(Y|\theta^d, X)$ for $X \to Y \leftarrow \boldsymbol{\theta}$, and $P_M^d(X|\hat{\theta}^d, Y) = P(X|\theta^d, Y)$ for $Y \to X \leftarrow \boldsymbol{\theta}$. If the LLM $M$ is invertible, then $\hat{\theta}^d = \theta^d$.*

*Proof.* The proof of this proposition is straightforward.

Since

$$\mathcal{L}(\hat{\theta}^d) = H(P(Y|\theta^d, X)) + KL(P(Y|\theta^d, X)||P_M^d(Y|\hat{\theta}^d, X))$$

when $\mathcal{L}(\hat{\theta}^d)$ is minimized, we have $P_M^d(Y|\hat{\theta}^d, X) = P(Y|\theta^d, X)$ for $X \to Y \leftarrow \boldsymbol{\theta}$, and $P_M^d(X|\hat{\theta}^d, Y) = P(X|\theta^d, Y)$ for $Y \to X \leftarrow \boldsymbol{\theta}$.

If $M$ is invertible, since the embedding matrix is invertible with or without new concept tokens, $P_M^d(Y|\hat{\theta}, X) = P_M^d(Y|\hat{\theta}', X)$ implies that $\hat{\theta} = \hat{\theta}'$. Thus $\boldsymbol{\theta}$ is identifiable, which means $\hat{\theta}^d = \theta^d$.
$\square$

Table 2: Prompt template and label mapping for the datasets we use. Since almost all sentences from ETHOS contain offensive content, we mask out the key offensive words in the examples below.

| Dataset | Prompt | Label Mapping |
|---|---|---|
| SST-2 | sentence: well worth revisiting as many times
positive | negative/positive |
| FPB | The company anticipates its turnover for the whole 2010 to surpass that of the previous year when it was EUR 67.1 million .
positive | negative/neutral/positive |
| COLA | It is this hat that I know the boy who is wearing.
unacceptable | acceptable/unacceptable |
| DBPedia | The Nucet River is a tributary of the Chiojdeanca River in Romania.
NaturalPlace | Album/Animal/Artist/
Athlete/Building/Company/
EducationalInstitution/Film/
MeanOfTransportation/
NaturalPlace/OfficeHolder/
Plant/Village/WrittenWork |
| EmoC | fast i mean fastingis a way of skipping meals i mena you move on too fast
others | angry/happy/others/sad |
| EmoS | i feel this place was tragic
sadness | anger/fear/joy/love/
sadness/surprise |
| ETHOS-SO | [Masked] should be removed from the face of the earth
true | false/true |
| ETHOS-R | I hate being a [Masked], wish I was a [Masked] and no [Masked] on earth existed
false | false/true |

## B  Experiments

**Dateset.** In Table 2, we show how we process the text classification datasets into prompts. For each dataset, we take at most 16384 examples from the training set for training, and uniformly sample at most 1000 examples from the test set to test the in-context learning performance. In Table 3, we show the train size and test size we used for each dataset. We also list the set of diverse tasks trained with each dataset, which are denoted by their name in Huggingface datasets.[9] The license for SST2, ETHOS-SO and ETHOS-R is GNU General Public License v3. FPB is under a Creative Commons Attribution-NonCommercial-ShareAlike 3.0 Unported License. Note that these two datasets are hate speech detection datasets for different kinds of hate speech and contain many offensive texts. COLA is excerpted from the published works available on the website, and the copyright (where applicable) remains with the original authors or publishers. DBpedia is under a Creative Commons Attribution-ShareAlike License and the GNU Free Documentation License. EmoC and EmoS should be used for educational and research purposes only.

**Experiment details.** We run our experiments on A100, V100, and A6000 GPUs. We adopt a large portion of the code from the MetaICL repository [25][10]. The training takes around 20 to 40 hours on a single GPU. We use a learning rate of 1e-4 and a batch size of 16, and train for 10k steps in total.

**Main results.** In Table 4, we list the detailed results of our method and baselines with different LLMs on different datasets in Figure 2.

**Causal direction results.** The detailed results with anti-causal direction (the opposite direction to what we described in Section 4 are in Table 7) are shown in Table 7, corresponding to Figure 6 in the main text.

**Other LLMs results.** The detailed results with other LLMs are shown in Table 6, corresponding to Figure 3a in the main text.

**Random token results.** The detailed results with random tokens are shown in Table 5, corresponding to Figure 3b in the main text.

---

[9]https://huggingface.co/docs/datasets/index
[10]https://github.com/facebookresearch/MetaICL

| datset $d$ | train size | test size | task set $\mathcal{S}$ |
|---|---|---|---|
| SST2 (glue-sst2) | 16384 | 1000 | glue-cola/glue-mnli/glue-qqp/
glue-mrpc/glue-qnli/glue-rte/glue-sst2/glue-wnli |
| FPB (financial_phrasebank) | 1811 | 453 | glue-sst2/glue-mnli/math_qa/sciq/
social_i_qa/wino_grande/glue-qqp/
ag_news/financial_phrasebank/
poem_sentiment/anli/quarel/quartz/
medical_questions_pairs/paws/dbpedia_14 |
| COLA (cola-sst2) | 8551 | 1000 | glue-cola/glue-mnli/glue-qqp/glue-mrpc/
glue-qnli/glue-rte/glue-sst2/glue-wnli |
| DBpedia (dbpedia_14) | 16384 | 1000 | glue-sst2/glue-mnli/math_qa/sciq/
social_i_qa/wino_grande/glue-qqp/
ag_news/financial_phrasebank/
poem_sentiment/anli/quarel/quartz/
medical_questions_pairs/paws/dbpedia_14 |
| EmoC (emo) | 16384 | 1000 | glue-sst2/amazon_polarity/
financial_phrasebank/poem_sentiment/
yelp_polarity/glue-cola/blimp/ag_news/
dbpedia_14/ethos/emo/emotion |
| EmoS (emotion) | 16000 | 1000 | glue-sst2/amazon_polarity/
financial_phrasebank/poem_sentiment/
yelp_polarity/glue-cola/blimp/ag_news/
dbpedia_14/ethos/emo/emotion |
| ETHOS-SO (ethos-sexual_orientation) | 346 | 87 | glue-sst2/amazon_polarity/
financial_phrasebank/poem_sentiment/
yelp_polarity/glue-cola/blimp/ag_news/
dbpedia_14/ethos/emo/emotion |
| ETHOS-R (ethos-religion) | 346 | 87 | glue-sst2/amazon_polarity/
financial_phrasebank/poem_sentiment/
yelp_polarity/glue-cola/blimp/ag_news/
dbpedia_14/ethos/emo/emotion |

Table 3: Dataset details

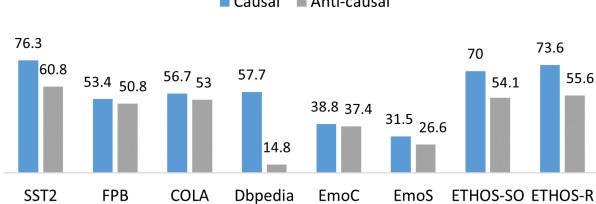

Figure 6: Accuracy of randomly selected demonstrations averaged over seven different LLMs except for GPT3-davinci, using the adopted *causal* direction and the *anti-causal* direction.

$k$**-ablation study results.** The detailed results of $k$ ablation study are shown in Table 10, corresponding to Figure 4a in the main text. In this experiment, we do not reorder the selected demonstrations according to Equation (3), as we need to use GPT2-large for the reordering, and it cannot fit in all the demonstrations. Instead, we order the selected demonstrations from the largest $\hat{P}_M^d(\theta^d|X^d, Y^d)$ to the smallest.

$c$**-ablation study results.** The detailed results of $c$ ablation study are shown in Table 11, corresponding to Figure 4b in the main text.

**Effect of using ground truth labels.** According to [26], the ground truth label is not necessary for demonstrations to have a good in-context learning performance, which we found is not entirely true for all the tasks. We compare our method with the randomly selected demonstration baseline under three scenarios: (a) **Original**: demonstrations with the correct labels; (b) **Random words**: using a random label projection map $\tau^d$ instead of a meaningful one. i.e., map each label to a fixed random word. In this case, the mapping from the input tokens $X$ to the labels $Y$ is still preserved; (c) **Random labels**: assign a random label to each demonstration, with the original label projection map $\tau^d$. As shown in Figure 7, by using a random label projection map or randomly assigning the labels, the performance of the randomly selected demonstration baseline drops considerably. And randomize the label assignment gives a larger performance drop than only using a random label projection map,

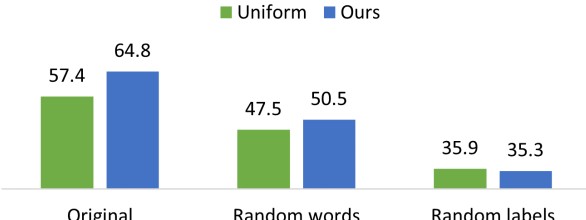

Figure 7: In-context learning accuracy of our method versus random selection baseline, with (a) ground truth labels (*original*), (b) random label mapping (*random words*), or random label assignments (*random label*), averaged over all eight datasets. Numbers are obtained with GPT2-large.

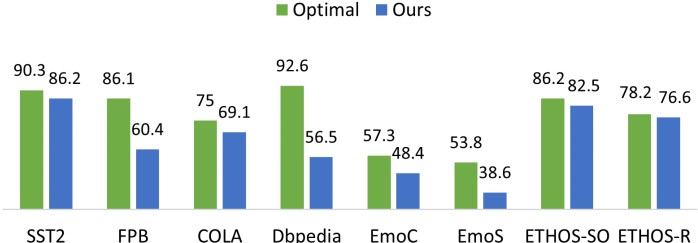

Figure 8: Accuracy of in-context learning using our method versus the theoretical maximum accuracy obtained using the learned concept tokens as prefixes. Numbers are obtained with GPT2-large.

which shows that the mapping between $X$ and $Y$ in the demonstrations matters. This indicates that in-context learning infers the mapping between $X$ and $Y$ from the demonstrations instead of merely invoking some learned function stored in the LLM parameters based on the appearance of $X$ and $Y$. We also show that the demonstrations selected by our method represent the $X - Y$ mapping better, as under the **Random words** condition, our method performs better than the random selection baseline, while our method does not improve the random selection baseline under the **Random labels** condition. The detailed results with random words and random labels are shown in Table 8

**Optimal performance** As stated in Theorem 2.3, the optimal performance of an in-context learning classifier is the Bayes optimal classifier $\arg\max_{y\in\mathcal{Y}} P_M^d(Y = y|\theta^d, X)$, which is approximated by using the learned concept tokens as prefixes. Note that this approximated Bayes optimal classifier cannot be transferred across different LLMs, as the learned concept tokens embeddings are aligned with a specific LLM. The advantage of in-context learning with our method is that the demonstrations can be transferred to any LLMs without training. Here we only compare the accuracy of in-context learning with our method and the approximated Bayes optimal classifier using GPT2-large, as it is the LLM that concept tokens are fine-tuned with. As shown in Figure 8, our method comes close to the optimal accuracy on many datasets, while there are some datasets that our method is lagging. This indicates that there are two ways to improve our method: the first is to improve the performance of the optimal classifier, by introducing a better latent concept learning algorithm. The other way is to reduce the performance gap between our method and the optimal classifier, by improving the demonstration selection algorithm. The detailed results using the learned concept tokens as prefixes are shown in Table 9.

**Reordering results.** We reorder the selected demonstrations to maximize the posterior of the concept tokens:

$$\arg\max_{\pi\in\Pi} \hat{P}_M^d(\theta^d|\pi((X_1^d, Y_1^d), ..., (X_k^d, Y_k^d)))\tag{3}$$

Where $\pi((X_1^d, Y_1^d), ..., (X_k^d, Y_k^d))$ is a permutation of $(X_1^d, Y_1^d), ..., (X_k^d, Y_k^d)$. $\Pi$ is the set of all possible permutations of the $k$ demonstrations. The detailed results with and without reordering are shown in Table 12, corresponding to Figure 9.

**Similar tokens.** We show the top ten similar tokens to some learned concept tokens in Table 13, as summarized in Figure 5 in the main text.

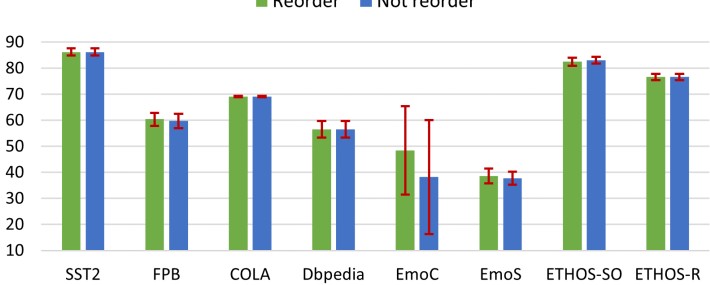

Figure 9: In-context learning accuracy of our method versus random selection baseline, with and without reordering. The red error bars represent the standard deviation across five runs. Numbers are obtained with GPT2-large.

Table 4: Accuracy of selected demonstration. Our demonstrations are selected using GPT2-large, and the same set of demonstrations is applied to all different LLMs. All LLMs are pre-trained only with the language modeling objective, while the pre-training data size of GPT2s is much smaller than GPT3s.

| LLM | Method | SST2 | FPB | COLA | DBpedia | EmoC | EmoS | ETHOS-SO | ETHOS-R | Avg |
|---|---|---|---|---|---|---|---|---|---|---|
| GPT2 | Uniform | 69.7 ± 1.8 | 52.9 ± 2.3 | 61.9 ± 1.4 | 48.0 ± 0.7 | 35.3 ± 1.7 | 26.4 ± 1.0 | 64.1 ± 4.8 | 71.0 ± 1.8 | 53.7 |
| (124M) | Similar | 69.5 ± 0.6 | 55.9 ± 1.7 | 63.2 ± 1.2 | 44.7 ± 3.1 | 36.4 ± 2.0 | 26.6 ± 1.3 | 77.7 ± 2.7 | 80.0 ± 3.7 | 56.8 |
|  | **Ours** | 76.8 ± 2.9 | 64.5 ± 3.2 | 69.1 ± 0.2 | 53.5 ± 2.95 | 37.2 ± 11.1 | 30.6 ± 4.8 | 80.9 ± 1.9 | 76.8 ± 2.6 | 61.2 |
| GPT2-m | Uniform | 70.8 ± 1.3 | 52.0 ± 1.7 | 57.8 ± 1.3 | 49.3 ± 2.0 | 34.2 ± 1.8 | 34.2 ± 1.8 | 76.3 ± 4.9 | 74.7 ± 2.2 | 56.2 |
| (355M) | Similar | 75.0 ± 1.9 | 57.7 ± 2.0 | 57.5 ± 2.2 | 47.9 ± 6.0 | 37.2 ± 3.6 | 35.2 ± 1.8 | 86.9 ± 2.9 | 84.6 ± 4.3 | 60.3 |
|  | **Ours** | 81.2 ± 1.3 | 59.3 ± 4.3 | 69.0 ± 0.2 | 52.9 ± 2.3 | 40.4 ± 21.5 | 37.2 ± 2.4 | 83.7 ± 1.1 | 76.8 ± 1.1 | 62.6 |
| GPT2-l | Uniform | 77.1 ± 1.2 | 51.3 ± 2.4 | 62.7 ± 0.8 | 54.4 ± 0.9 | 38.7 ± 2.1 | 34.5 ± 1.2 | 67.6 ± 4.3 | 72.9 ± 2.8 | 57.4 |
| (774M) | Similar | 80.7 ± 1.6 | 54.8 ± 3.8 | 50.9 ± 1.4 | 51.1 ± 5.2 | 39.9 ± 2.6 | 35.1 ± 2.1 | 80.9 ± 2.8 | 84.4 ± 2.6 | 59.7 |
|  | **Ours** | 86.2 ± 1.4 | 60.4 ± 2.5 | 69.1 ± 0.2 | 56.5 ± 3.2 | 48.4 ± 17.0 | 38.6 ± 2.8 | 82.5 ± 1.5 | 76.6 ± 1.2 | 64.8 |
| GPT2-xl | Uniform | 74.7 ± 0.9 | 53.2 ± 1.9 | 55.8 ± 1.6 | 53.0 ± 1.9 | 38.2 ± 1.5 | 38.2 ± 1.5 | 67.8 ± 6.4 | 72.6 ± 4.1 | 56.7 |
| (1.5B) | Similar | 80.6 ± 1.3 | 53.0 ± 2.5 | 55.0 ± 2.5 | 51.6 ± 5.9 | 39.9 ± 2.0 | 32.9 ± 2.1 | 82.8 ± 2.2 | 83.9 ± 4.5 | 60 |
|  | **Ours** | 83.1 ± 3.6 | 62.0 ± 2.5 | 68.9 ± 0.2 | 58.6 ± 3.3 | 43.6 ± 16.4 | 43.6 ± 16.4 | 83.0 ± 1.3 | 77.9 ± 1.3 | 65.1 |
| GPT3-a | Uniform | 76.9 ± 0.7 | 56.6 ± 1.1 | 53.1 ± 1.8 | 62.1 ± 1.4 | 38.6 ± 1.4 | 27.7 ± 1.3 | 65.5 ± 5.7 | 74.0 ± 3.0 | 56.8 |
| (350M) | Similar | 78.7 ± 1.0 | 52.2 ± 2.7 | 53.1 ± 1.8 | 54.6 ± 1.7 | 42.4 ± 3.5 | 37.2 ± 1.1 | 84.1 ± 2.2 | 87.8 ± 3.5 | 61.3 |
|  | **Ours** | 85.4 ± 1.7 | 61.9 ± 10.5 | 58.2 ± 7.0 | 64.0 ± 4.4 | 43.0 ± 7.2 | 37.9 ± 2.3 | 84.4 ± 1.4 | 78.9 ± 0.9 | 64.2 |
| GPT3-b | Uniform | 80.8 ± 0.6 | 55.2 ± 3.3 | 46.8 ± 2.0 | 66.5 ± 1.4 | 42.0 ± 0.7 | 27.0 ± 1.2 | 71.0 ± 4.6 | 72.6 ± 3.1 | 57.7 |
| (1.3B) | Similar | 83.9 ± 1.3 | 56.2 ± 2.3 | 45.1 ± 1.8 | 59.8 ± 1.8 | 42.9 ± 3.5 | 38.1 ± 1.7 | 86.7 ± 3.0 | 86.4 ± 3.0 | 62.4 |
|  | **Ours** | 87.3 ± 2.0 | 64.3 ± 5.9 | 67.2 ± 0.9 | 70.2 ± 3.2 | 43.6 ± 13.0 | 38.9 ± 5.0 | 84.6 ± 0.9 | 78.9 ± 1.2 | 66.9 |
| GPT3-c | Uniform | 84.2 ± 1.4 | 52.6 ± 1.8 | 59.1 ± 1.5 | 70.6 ± 0.8 | 44.3 ± 2.5 | 32.3 ± 1.9 | 77.5 ± 4.7 | 77.5 ± 0.6 | 62.3 |
| (6.7B) | Similar | 85.7 ± 1.4 | 62.2 ± 0.9 | 58.0 ± 1.7 | 62.2 ± 2.0 | 47.4 ± 4.3 | 39.8 ± 1.7 | 89.2 ± 1.4 | 89.7 ± 1.9 | 66.8 |
|  | **Ours** | 88.8 ± 0.7 | 64.1 ± 5.7 | 69.0 ± 0.3 | 73.6 ± 2.9 | 50.3 ± 11.9 | 43.1 ± 4.6 | 86.2 ± 0.0 | 78.2 ± 0.0 | 69.2 |
| GPT3-d | Uniform | 86.5 ± 0.9 | 59.2 ± 2.4 | 45.5 ± 2.8 | 73.6 ± 1.9 | 39.4 ± 0.7 | 40.6 ± 1.7 | 77.2 ± 2.6 | 76.8 ± 3.5 | 62.4 |
| (175B) | Similar | 88.5 ± 0.8 | 55.4 ± 3.3 | 45.4 ± 1.5 | 67.2 ± 1.8 | 37.6 ± 1.6 | 39.8 ± 1.4 | 86.9 ± 2.4 | 89.0 ± 3.8 | 63.7 |
|  | **Ours** | 87.8 ± 3.4 | 62.7 ± 3.3 | 58.5 ± 8.2 | 75.5 ± 2.4 | 41.3 ± 3.6 | 42.7 ± 3.9 | 85.1 ± 0.0 | 79.3 ± 0.0 | 66.6 |
| Avg | Uniform | 77.6 | 54.1 | 55.3 | 59.7 | 38.8 | 32.6 | 70.9 | 74.0 | 57.9 |
|  | Similar | 80.3 | 55.9 | 53.5 | 54.9 | 40.5 | 35.6 | 84.4 | 85.7 | 61.4 |
|  | **Ours** | 84.6 | 62.4 | 66.1 | 63.1 | 43.5 | 39.1 | 83.8 | 77.9 | 65.0 |

**Likelihood histogram.** We also show histograms of the probability of each example predicting corresponding concept tokens in different datasets. We can see that the probability of prediction concept tokens can well differentiate examples in a dataset.

**Selected demonstrations.** Table 14 shows the selected top 4 demonstration by our proposed algorithm.

| Task | Selected demonstrations |
|---|---|

| | |
|---|---|
| GSM8K | Question: It takes Bryan 5 minutes to walk from his house to the bus station. Then he rides the bus for 20 minutes. After that, he walks 5 minutes from the bus station to his job. It takes the same amount of time in the morning and the evening. How many hours per year does Bryan spend traveling to and from work, if he works every day? Bryan spends 5+20+5 =«5+20+5=30»30 minutes traveling to work. He does this twice a day, so he spends 30*2=«30*2=60»60 minutes traveling per day. Thus, he spends 60/60=«60/60=1»1 hour traveling to and from work every day. Thus, he spends 1 hour*365 days =«1*365=365»365 hours traveling per year. The answer is: 365 |
| GSM8K | Question: Cherry put up a delivery service. She charges $2.50 for a 3-5 kilograms cargo and $4 for a 6-8 kilograms cargo. If she delivers four 5 kilograms cargo and two 8 kilograms cargo per day, how much money will she earn in a week? Cherry earns $2.50 x 4= $«2.5*4=10»10 after delivering four 5 kilograms of cargo per day. She earns $4 x 2= $«4*2=8»8 after delivering two 8 kilograms of cargo per day. So, her total earning per day is $8+$10= $«8+10=18»18. Therefore, she will earn $18 x 7= $«18*7=126»126 in a week. The answer is: 126 |
| GSM8K | Question: Bill is laying power cable for a new neighborhood. There are going to be 18 east-west streets that are 2 miles long and 10 north-south streets that are four miles long. It takes 5 miles of cable to electrify 1 mile of street. If cable costs $2000/mile, what is the total cost of cable for the neighborhood? First find the total distance of the east-west streets: 18 streets * 2 miles/street = «18*2=36»36 miles. Then find the total distance of the north-south streets: 10 streets * 4 miles/street = «10*4=40»40 miles. Then add the number of miles from each type of street to find the total distance: 36 miles + 40 miles = «36+40=76»76 miles. Then multiply that number by 5 to find the number of miles of cable needed: 76 miles street * 5 miles cable/mile street = «76*5=380»380 miles of cable. Then multiply that number by the cost of one mile of cable to find the total cost: 380 miles * $2000/mile = $«380*2000=760000»760,000. The answer is: 760000 |
| GSM8K | Question: John buys a gaming PC for $1200. He decides to replace the video card in it. He sells the old card for $300 and buys a new one for $500. How much money did he spend on his computer, counting the savings from selling the old card? He spent an extra 500-300=$«500-300=200»200 on the video card. That means the total cost was 1200+200=$«1200+200=1400»1400. The answer is: 1400 |
| SST2 | sentence: faced and spindly attempt at playing an ingenue makes her nomination as best actress even more of a an a positive |
| SST2 | sentence: holofcener's film offers just enough insight to keep it from being simpleminded, and positive |
| SST2 | sentence: i'm not a fan of the phrase ' life affirming' because it usually means ' schmaltzy,' but real women have curves truly is life affirming negative |

| | |
|---|---|
| SST2 | sentence: the script is about as interesting as a recording of conversations at the wal-mart checkout line negative |
| DBpedia | OfficeHolder Lucie Papin (born September 7 1936) is a former Canadian politician who served in both the House of Commons and Senate. |
| DBpedia | Village Kunkalamarru is very renowned village under Karamchedu Mandal which is located about 15 km from the busy commercial town of Chirala in Prakasam district in the state of Andhra Pradesh India.Its neighbouring villages are Karamchedu Veerannapalem. |
| DBpedia | EducationalInstitution The Pontifical Catholic University of Puerto Rico at Mayagez is a university located in the city of Mayagez Puerto Rico. It is part of the Pontifical Catholic University of Puerto Rico. The university began as an extension of the Catholic University of Puerto Rico in the early 1960s. In 1982 it was awarded the official title of Center and later it became the Mayagez Campus of the Pontifical Catholic University of Puerto Rico at Mayagez in 1996. |
| DBpedia | Artist Choi Dong-wook [citation needed]; born November 9 1984) better known by his stage name Se7en is a South Korean singer from YG Entertainment. He has also advanced into Japan China and the United States. |

Table 14: Selected demonstrations by our method.

## C  Limitations and Future Work

While the assumption that a large language model captures the true distribution of language is fairly common in the literature studying LLMs [50, 34], this assumption is not entirely accurate in practice. According to [15], LLMs systematically underestimate rare text sequences, which constitute a significant portion of the long-tail distribution of language. Although this assumption is adequate to achieve favorable empirical results, it is expected that more accurate language models will, in theory, lead to improved outcomes.

The selection of the accompanying diverse tasks $\mathcal{S}$ is currently left to the user's discretion. A better approach to constructing such a task set is needed to gain a deeper understanding of latent concept variables and to improve the latent concept learning algorithm.

Our algorithm currently only applies to classification tasks. More complex latent variables could be designed to improve the in-context learning performance of more complex tasks like math word questions and logical reasoning problems.

## D  Broader Impact

The utilization of language models (LLMs) for specific tasks is often hindered by the high cost associated with training or fine-tuning them. However, the in-context learning paradigm offers a cost-effective and convenient alternative for utilizing the power of pre-trained LLMs. Our work has demonstrated a significant improvement in the performance of in-context learning through a relatively low-cost and simple approach, thus making the use of LLMs more accessible for individuals with limited resources.

However, it is important to consider the broader implications of the increasing use of LLMs. As LLMs are not infallible and may make mistakes, it is crucial to explicitly warn users of the potential for misleading output and to regulate the distribution of LLMs in order to prevent any negative societal impact. Additionally, it is possible that LLMs could be intentionally misused, thus it is important to consider the ethical implications of their use and to take appropriate measures to mitigate

Table 5: Accuracy of selected demonstration. Our demonstrations are selected using GPT2-large, and the same set of demonstrations is applied to all different LLMs. All LLMs are pre-trained only with the language modeling objective, while the pre-training data size of GPT2s is much smaller than GPT3s.

| LLM | Method | SST2 | FPB | COLA | DBpedia | EmoC | EmoS | ETHOS-SO | ETHOS-R | Avg |
|---|---|---|---|---|---|---|---|---|---|---|
| GPT2 | Uniform | 69.7 ± 1.8 | 52.9 ± 2.3 | 61.9 ± 1.4 | 48.0 ± 0.7 | 35.3 ± 1.7 | 26.4 ± 1.0 | 64.1 ± 4.8 | 71.0 ± 1.8 | 53.7 |
| (124M) | Random | 69.8 ± 3.3 | 51.1 ± 1.7 | 69.0 ± 0.1 | 49.0 ± 4.5 | 33.7 ± 15.5 | 24.2 ± 7.6 | 66.4 ± 17.5 | 66.2 ± 16.2 | 53.7 |
|  | **Ours** | 76.8 ± 2.9 | 64.5 ± 3.2 | 69.1 ± 0.2 | 53.5 ± 2.95 | 37.2 ± 11.1 | 30.6 ± 4.8 | 80.9 ± 1.9 | 76.8 ± 2.6 | 61.2 |
| GPT2-l | Uniform | 77.1 ± 1.2 | 51.3 ± 2.4 | 62.7 ± 0.8 | 54.4 ± 0.9 | 38.7 ± 2.1 | 34.5 ± 1.2 | 67.6 ± 4.3 | 72.9 ± 2.8 | 57.4 |
| (774M) | Random | 81.9 ± 4.5 | 46.5 ± 4.7 | 64.9 ± 7.8 | 50.3 ± 4.3 | 42.5 ± 16.7 | 36.1 ± 6.5 | 67.6 ± 20.4 | 67.8 ± 15.0 | 57.2 |
|  | **Ours** | 86.2 ± 1.4 | 60.4 ± 2.5 | 69.1 ± 0.2 | 56.5 ± 3.2 | 48.4 ± 17.0 | 38.6 ± 2.8 | 82.5 ± 1.5 | 76.6 ± 1.2 | 64.8 |

Table 6: We test our method on other similar sizes (6-7B) LLMs.

| LLM | Method | SST2 | FPB | COLA | DBpedia | EmoC | EmoS | ETHOS-SO | ETHOS-R | Avg |
|---|---|---|---|---|---|---|---|---|---|---|
| GPT2-l | Random | 77.1 ± 1.2 | 51.3 ± 2.4 | 62.7 ± 0.8 | 54.4 ± 0.9 | 38.7 ± 2.1 | 34.5 ± 1.2 | 67.6 ± 4.3 | 72.9 ± 2.8 | 57.4 |
|  | **Ours** | 86.2 ± 1.4 | 60.4 ± 2.5 | 69.1 ± 0.2 | 56.5 ± 3.2 | 48.4 ± 17.0 | 38.6 ± 2.8 | 82.5 ± 1.5 | 76.6 ± 1.2 | 64.8 |
| GPT3-c | Random | 84.2 ± 1.4 | 52.6 ± 1.8 | 59.1 ± 1.5 | 70.6 ± 0.8 | 44.3 ± 2.5 | 32.3 ± 1.9 | 77.5 ± 4.7 | 77.5 ± 0.6 | 62.3 |
|  | **Ours** | 88.8 ± 0.7 | 64.1 ± 5.7 | 69.0 ± 0.3 | 73.6 ± 2.9 | 50.3 ± 11.9 | 43.1 ± 4.6 | 86.2 ± 0.0 | 78.2 ± 0.0 | 69.2 |
| GPT-J | Random | 78.5 ± 1.0 | 53.1 ± 1.7 | 58.3 ± 2.2 | 55.6 ± 1.2 | 38.5 ± 2.0 | 33.3 ± 1.5 | 76.6 ± 3.7 | 76.6 ± 1.4 | 58.8 |
|  | **Ours** | 87.8 ± 1.9 | 56.7 ± 4.3 | 69.1 ± 0.2 | 60.0 ± 3.6 | 32.5 ± 16.1 | 33.2 ± 2.8 | 85.3 ± 0.5 | 77.0 ± 0.0 | 62.7 |
| OPT | Random | 72.4 ± 0.8 | 32.8 ± 0.3 | 34.8 ± 0.6 | 29.4 ± 1.4 | 67.1 ± 1.8 | 36.9 ± 0.6 | 86.2 ± 0.0 | 78.2 ± 0.0 | 54.7 |
|  | **Ours** | 74.2 ± 3.0 | 34.1 ± 6.1 | 35.7 ± 3.1 | 28.8 ± 2.1 | 76.7 ± 4.1 | 39.0 ± 3.4 | 86.2 ± 0.0 | 78.2 ± 0.0 | 56.6 |
| LLaMA | Random | 57.7 ± 1.5 | 23.7 ± 1.3 | 30.8 ± 0.2 | 15.8 ± 0.8 | 4.4 ± 0.7 | 35.2 ± 0.7 | 66.2 ± 5.8 | 57.2 ± 5.1 | 36.4 |
|  | **Ours** | 60.5 ± 4.7 | 19.1 ± 1.9 | 30.8 ± 0.2 | 16.9 ± 1.3 | 4.3 ± 0.7 | 35.3 ± 0.6 | 77.2 ± 13.6 | 56.3 ± 10.8 | 37.6 |

any potential negative effects. We posit that these regulations and measures should be put in place at the time of distributing LLMs to ensure the safe and responsible use of these models. Furthermore, as we publicly release our code, we will also provide clear warnings and guidelines to users to ensure that the potential risks associated with the use of our method are fully understood and addressed.

Table 7: We test random selection baseline with anti-causal direction.

| LLM | SST2 | FPB | COLA | DBpedia | EmoC | EmoS | ETHOS-SO | ETHOS-R |
|---|---|---|---|---|---|---|---|---|
| GPT2 | $57.4 \pm 1.9$ | $56.6 \pm 2.1$ | $55.9 \pm 1.7$ | $11.3 \pm 1.0$ | $24.6 \pm 2.4$ | $22.1 \pm 1.1$ | $64.1 \pm 4.8$ | $58.6 \pm 5.5$ |
| GPT2-m | $56.7 \pm 1.6$ | $48.7 \pm 2.1$ | $55.3 \pm 1.8$ | $13.9 \pm 1.2$ | $22.4 \pm 1.9$ | $24.9 \pm 2.3$ | $44.8 \pm 1.9$ | $45.5 \pm 3.5$ |
| GPT2-l | $58.7 \pm 0.7$ | $33.7 \pm 1.3$ | $50.8 \pm 1.6$ | $13.6 \pm 1.3$ | $28.2 \pm 3.6$ | $26.2 \pm 2.7$ | $48.7 \pm 3.7$ | $53.6 \pm 5.3$ |
| GPT2-xl | $54.2 \pm 0.5$ | $46.8 \pm 1.2$ | $50.6 \pm 1.1$ | $12.6 \pm 1.5$ | $31.4 \pm 2.8$ | $25.9 \pm 3.2$ | $65.5 \pm 4.9$ | $61.8 \pm 1.5$ |
| GPT3-a | $55.8 \pm 0.9$ | $58.9 \pm 2.1$ | $51.6 \pm 1.4$ | $14.3 \pm 0.8$ | $54.2 \pm 3.1$ | $27.7 \pm 1.3$ | $49.2 \pm 3.3$ | $54.9 \pm 6.4$ |
| GPT3-b | $64.4 \pm 1.6$ | $58.9 \pm 2.6$ | $53.4 \pm 1.1$ | $14.6 \pm 1.1$ | $52.0 \pm 2.5$ | $27.0 \pm 1.3$ | $48.3 \pm 2.7$ | $51.0 \pm 4.0$ |
| GPT3-c | $78.2 \pm 1.6$ | $52.3 \pm 2.3$ | $53.7 \pm 0.7$ | $23.0 \pm 2.5$ | $49.1 \pm 2.6$ | $32.2 \pm 1.9$ | $57.9 \pm 2.7$ | $64.1 \pm 5.0$ |
| Avg | 60.8 | 50.8 | 53 | 14.8 | 37.4 | 26.6 | 54.1 | 55.6 |

Table 8: We test our method with random words and random labels using GPT2-large.

| | Method | SST2 | FPB | COLA | DBpedia | EmoC | EmoS | ETHOS-SO | ETHOS-R | Avg |
|---|---|---|---|---|---|---|---|---|---|---|
| R words | Random | $54.1 \pm 4.2$ | $43.4 \pm 1.9$ | $62.2 \pm 4.9$ | $11.2 \pm 0.9$ | $32.4 \pm 5.2$ | $19.1 \pm 1.8$ | $80.7 \pm 4.8$ | $77.0 \pm 3.6$ | 47.5 |
| | **Ours** | $50.3 \pm 1.3$ | $44.9 \pm 4.2$ | $69.2 \pm 0.2$ | $13.9 \pm 1.2$ | $37.8 \pm 12.1$ | $23.5 \pm 7.4$ | $86.0 \pm 0.5$ | $77.9 \pm 0.5$ | 50.5 |
| R labels | Random | $51.5 \pm 0.9$ | $32.5 \pm 1.2$ | $49.3 \pm 3.0$ | $6.7 \pm 1.0$ | $25.1 \pm 0.6$ | $17.2 \pm 0.9$ | $48.0 \pm 2.5$ | $56.8 \pm 3.1$ | 35.9 |
| | **Ours** | $49.6 \pm 0.9$ | $36.2 \pm 2.5$ | $49.3 \pm 1.6$ | $6.6 \pm 0.2$ | $24.7 \pm 0.6$ | $16.6 \pm 1.0$ | $51.0 \pm 4.9$ | $48.7 \pm 3.5$ | 35.3 |

Table 9: Accuracy using concept tokens as prefixes.

| SST2 | FPB | COLA | DBpedia | EmoC | EmoS | ETHOS-SO | ETHOS-R |
|---|---|---|---|---|---|---|---|
| $90.3 \pm 0.0$ | $86.1 \pm 0.0$ | $75.0 \pm 0.1$ | $92.6 \pm 0.6$ | $57.3 \pm 1.8$ | $53.8 \pm 0.7$ | $86.2 \pm 0.0$ | $78.2 \pm 0.0$ |

Table 10: $k$ ablation study using GPT2-large, without reordering.

| | Method | SST2 | FPB | COLA | DBpedia | EmoC | EmoS | ETHOS-SO | ETHOS-R | Avg |
|---|---|---|---|---|---|---|---|---|---|---|
| $k = 2$ | Random | $74.4 \pm 1.0$ | $48.5 \pm 1.1$ | $48.9 \pm 1.6$ | $52.9 \pm 2.0$ | $42.8 \pm 0.6$ | $37.1 \pm 1.2$ | $66.9 \pm 4.7$ | $66.4 \pm 6.8$ | 54.7 |
| | **Ours** | $78.1 \pm 4.5$ | $50.1 \pm 2.9$ | $54.3 \pm 8.8$ | $57.3 \pm 5.1$ | $41.1 \pm 9.8$ | $36.1 \pm 2.6$ | $84.6 \pm 1.6$ | $76.8 \pm 4.5$ | 59.8 |
| $k = 4$ | Random | $76.9 \pm 0.7$ | $56.6 \pm 1.1$ | $53.1 \pm 1.8$ | $62.1 \pm 1.4$ | $38.6 \pm 1.4$ | $27.7 \pm 1.3$ | $65.5 \pm 5.7$ | $74.0 \pm 3.0$ | 56.8 |
| | **Ours** | $86.2 \pm 1.4$ | $59.7 \pm 2.8$ | $69.1 \pm 0.2$ | $56.5 \pm 3.2$ | $38.2 \pm 21.8$ | $37.7 \pm 2.5$ | $83.0 \pm 1.3$ | $76.6 \pm 1.2$ | 63.4 |
| $k = 8$ | Random | $79.9 \pm 0.2$ | $57.1 \pm 1.6$ | $51.3 \pm 1.0$ | $66.5 \pm 1.2$ | $37.6 \pm 1.5$ | $36.2 \pm 0.6$ | $68.5 \pm 3.5$ | $72.9 \pm 3.3$ | 58.8 |
| | **Ours** | $87.0 \pm 2.4$ | $59.9 \pm 3.3$ | $55.3 \pm 9.7$ | $67.0 \pm 0.9$ | $39.9 \pm 5.3$ | $38.8 \pm 2.6$ | $77.0 \pm 11.1$ | $78.9 \pm 0.9$ | 63 |
| $k = 16$ | Random | $79.9 \pm 1.1$ | $54.9 \pm 2.7$ | $54.5 \pm 2.8$ | $69.1 \pm 1.1$ | $33.7 \pm 2.2$ | $33.5 \pm 1.4$ | $64.8 \pm 4.0$ | $69.0 \pm 3.2$ | 57.4 |
| | **Ours** | $84.6 \pm 1.9$ | $60.4 \pm 6.4$ | $62.0 \pm 7.0$ | $71.0 \pm 1.9$ | $37.2 \pm 6.1$ | $37.1 \pm 2.2$ | $72.4 \pm 7.6$ | $74.7 \pm 4.7$ | 62.4 |

Table 11: $c$ ablation study using GPT2-large

| | SST2 | FPB | COLA | DBpedia | EmoC | EmoS | ETHOS-SO | ETHOS-R | Avg |
|---|---|---|---|---|---|---|---|---|---|
| $c = 5$ | $78.9 \pm 2.4$ | $59.8 \pm 10.8$ | $34.3 \pm 5.0$ | $62.9 \pm 2.4$ | $44.9 \pm 9.5$ | $38.1 \pm 2.4$ | $71.7 \pm 5.9$ | $62.1 \pm 19.7$ | 56.6 |
| $c = 10$ | $85.4 \pm 1.7$ | $61.9 \pm 10.5$ | $58.2 \pm 7.0$ | $64.0 \pm 4.4$ | $43.0 \pm 7.2$ | $37.9 \pm 2.3$ | $84.4 \pm 1.4$ | $78.9 \pm 0.9$ | 64.2 |
| $c = 15$ | $80.1 \pm 1.4$ | $64.3 \pm 7.7$ | $63.1 \pm 9.4$ | $58.7 \pm 3.2$ | $36.4 \pm 11.5$ | $38.6 \pm 1.9$ | $80.9 \pm 3.9$ | $76.3 \pm 5.9$ | 62.3 |
| $c = 20$ | $78.5 \pm 4.1$ | $51.8 \pm 8.0$ | $66.5 \pm 2.3$ | $58.0 \pm 3.4$ | $36.3 \pm 4.3$ | $41.8 \pm 5.8$ | $80.7 \pm 4.5$ | $73.8 \pm 5.4$ | 60.92 |

Table 12: Reorder versus not reorder using our method, with GPT2-large.

| | SST2 | FPB | COLA | DBpedia | EmoC | EmoS | ETHOS-SO | ETHOS-R | Avg |
|---|---|---|---|---|---|---|---|---|---|
| reorder | $86.2 \pm 1.4$ | $60.4 \pm 2.5$ | $69.1 \pm 0.2$ | $56.5 \pm 3.2$ | $48.4 \pm 17.0$ | $38.6 \pm 2.8$ | $82.5 \pm 1.5$ | $76.6 \pm 1.2$ | 64.8 |
| not reorder | $86.2 \pm 1.4$ | $59.7 \pm 2.8$ | $69.1 \pm 0.2$ | $56.5 \pm 3.2$ | $38.2 \pm 21.8$ | $37.7 \pm 2.5$ | $83.0 \pm 1.3$ | $76.6 \pm 1.2$ | 63.4 |

Table 13: We list the top 10 similar words (tokens) to some of the learned concept tokens.

| concept token | similar words |
| --- | --- |
| FPB-2 | milo coordinate notify rendering benefiting routing EntityItem routed Messages Plot |
| FPB-3 | unlocked updating deleting dropping damage updates drops Gained taken dropped |
| FPB-4 | FX Safari Fixes advertisers Links Coins Operator marketers Guidelines |
| FPB-5 | 674 592 693 696 498 593 793 504 691 683 |
| COLA-1 | exha trunc curv fragmented elong iterator initialized bounds Iter filament |
| COLA-2 | Sp spa contributed cerv borrower paper tiger Erica USH Schwartz |
| COLA-7 | democr Barack WH ophobic neum Democrats Rachel WH Democrats |
| DBpedia-4 | often impede blockade incarcerated LEASE pollutants pesticides uphe lawmakers fossils |
| DBpedia-5 | categorized closes therapies antidepressant retrospective clinically physicians therapists randomized clinicians |
| DBpedia-7 | JS provided Killed richness Compet Nevertheless Probably Proceedings horizontally |
| ETHOS-SO-3 | Revolution Spread itu Million Pascal stabil Indy Georgian Figure resy |
| ETHOS-R-2 | council Chocobo Shant uyomi aditional cumbers subur ThumbnailImage araoh Pharaoh |
| ETHOS-R-8 | seems outlines emitted grin outline circuitry sized flips emits flipped |
| ETHOS-R-9 | 223 asel Cyrus Sith Scorpion Snape Jas Leia Ned Morty |
| EmoC-6 | behavi checkpoints unintention crib eleph looph np mosquit blat pione |
| EmoC-8 | depressed bullied choked stricken devastated unsuccessful cheated distraught troubled failing |
| EmoS-1 | frightened rebellious depressed careless bullied restless reluctant distraught clumsy disgruntled |
| EmoS-5 | obsessive crappy demonic delusions psychosis psychotic childish stupidity reckless insanity |
| EmoS-7 | benevolent charismatic perfected volunte unintention pione innocuous fearless glamorous ruthless |
| EmoS-9 | whispers pundits Sadly horribly curiously noticeably Sadly gaping painfully shockingly |

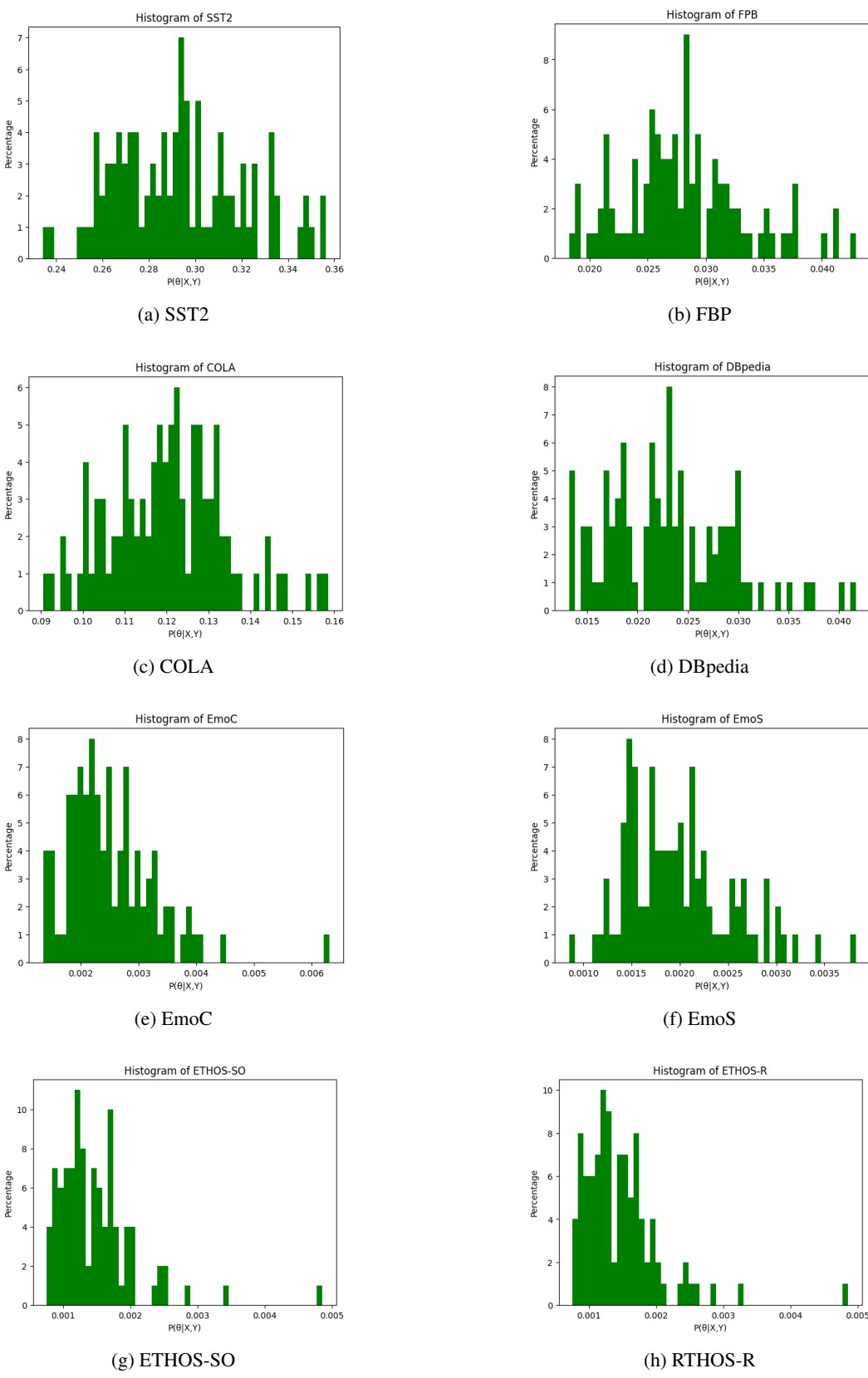

Figure 10: Historgrams of the probability of train examples in each dataset predicting corresponding concept tokens.

