# OpenReview forum: "Large Language Models Are Latent Variable Models: Explaining and Finding Good Demonstrations for In-Context Learning"
_NeurIPS.cc/2023/Conference — NeurIPS 2023 poster_

### Official Review · Reviewer_uvg2 · 2023-07-01

**Soundness:** 2 fair
**Presentation:** 3 good
**Contribution:** 2 fair
**Rating:** 5
**Confidence:** 4

**Summary:**

The paper presents a new perspective to understand the in-context learning behavior of large language models (LLMs) from the angle of latent concept learning resembling topic models. Based on the generative process defined by input data, latent concepts, and labels, the authors propose a two-stage algorithm to first learn the latent concept and then use it to select the best-performing demonstrations that boost in-context learning performance. Experiments on eight datasets show that the method is able to consistently outperform random selection and selection based on semantic similarity.

**Strengths:**

* Originality: Although the latent concept learning perspective is largely inspired by Xie et al., the goal and method proposed in this paper for selecting effective demonstrations is still sufficiently interesting and different from Xie et al., which seem novel to me.
* Quality: The paper first states its generative assumptions defined by input data, latent concepts, and labels, based on which it then derives a two-stage method to first learn the concept and then select demonstrations. The theories and methods seem solid to me.
* Clarity: The paper is overall clear and nicely written.
* Significance: While it's nice to see that the method is able to consistently outperform simple baselines across the tasks selected, I generally feel that the setting considered in this paper is somewhat artificial, and that the evaluation is not comprehensive enough to testify the generalization ability of the method. These concerns weaken the significance of the paper. See weaknesses below for details.

**Weaknesses:**

* Problem setting: The paper assumes access to a (relatively) large training set from which a few demonstrations can be drawn. With this amount of training data (e.g., 100), there could be better alternative choices than in-context learning. For example, one can tune an LLM with parameter-efficient methods and easily outperform in-context learning (shown in Liu et al.) without even having to consider how to select the best demonstrations. Of course, one can argue that in-context learning is applicable to non-open-source LLMs while training-based methods are not. However, given the recent growth in the availability of open-source LLMs such as LLaMA and Falcon, I believe it would be generally better to consider parameter-efficient tuning than in-context learning if the authors assume ~100 training samples are available. In summary, I feel that the problem setting of "carefully selecting good demonstrations from a larger training set" is somewhat artificial on its own from the very beginning.
* Evaluation tasks: The eight downstream tasks selected for evaluation appear to be too easy for nowadays LLMs, and it's unclear given the current evaluation how generalizable the method is to more challenging tasks like MMLU and reasoning, potentially when combined with chain-of-thought prompting.
* Typo: Line 104: "semantic analysis" -> "sentiment analysis"

Reference:
Liu et al. "Few-shot parameter-efficient fine-tuning is better and cheaper than in-context learning." NeurIPS (2022).

**Questions:**

* What is the role of $\boldsymbol{\epsilon}$ in the generative process (Line 91)? It seems like it is not taken into account in the current modeling process.
* How does the method work for instruction-tuned models (e.g., ChatGPT, Alpaca)?
* Why are the results of LLaMA that bad (Figure 3a)? LLaMA is generally recognized as a better-performing model than OPT series under similar sizes.

**Limitations:**

Please see the Weaknesses section.

---

> ### Author Rebuttal · Authors · 2023-08-09
>
> Thank you for your review. Below is our response to your comments:
>
> 1. **About how realistic is the problem setting**: We want first to reiterate that our problem setting is chosen to empirically prove the correctness of our proposed theory of in-context learning. We want to reiterate that improving and connecting a previously synthetic-only theory of in-context learning [6] with real-world models and data is already non-trivial and is relatively rare in the current theories of LLMs, which are usually disconnected from the real world. The value of our paper does not only lie in its real-world application. However, we do agree that we need to show real-world use cases of our proposed algorithm. In short, our proposed method is most useful with hard tasks that smaller LLMs perform significantly worse than larger models. Please refer to point 2 of our general response for results on GSM8K. In this case, even parameter-efficient fine-tuning with smaller models cannot obtain a reasonable performance (less than 4%), but in-context learning with both small and large models combined with our method can achieve much higher performance (19.3% using Llama2 7B and more than 80% using ChatGPT).
>
> 2. **About evaluation tasks**: Since our primary goal is to connect the theory with real-world models and datasets, we did not try to include harder tasks. However, to show that our method also works on more challenging tasks, we show new results on GSM8K in point 2 of our general response, which involved chain-of-thoughts reasoning and generation, and shows the 4-7.9% increase of using our method with Llama2 (7B) compared with the Unifrom baseline. We want to reiterate that improving and connecting a previously synthetic-only theory of in-context learning [6] with real-world models and data is already non-trivial and is relatively rare in the current theories of LLMs, which are usually disconnected from the real world. The value of our paper does not only lie in its real-world application.
>
> 3. **About $\epsilon$**: It represents a noise variable, with any zero-mean distribution. Without this variable, there is no randomness between $X$, $Y$, and $\theta$, as the functions $f$ and $g$ are all deterministic. The $\epsilon$ here is just to introduce some noises such that the conditional distribution $P(Y|X, \theta)$ is meaningful. Thanks for pointing this out. We will clarify this in the revision.
>
> 4. **About instruction-tuned models**: In point 2 of our general response, we include results with ChatGPT on GSM8K, which shows our proposed method also works on instruction-tuned models (Accuracy of random selection baseline: 76.5%; Ours: 81.2%).
>
> 5. **About Llama's performance**: The bad performance of Llama is also surprising to us. We wrote an email to the authors of Llama to ask about this, and they replied that they had never tested under such a scenario, so they don’t know. The code for testing Llama is completely the same as for testing other models, though. Our understanding is that Llama (first generation) is good at generation tasks, but not so good at in-context learning with simple classification tasks.

---

> > ### Comment · Reviewer_uvg2 · 2023-08-19
> >
> > I thank the authors for their response. I'm still relatively positive about the paper and I'm keeping my original rating.

---

> > > ### Author Response · Authors · 2023-08-19
> > >
> > > Thank you for taking the time to read our rebuttal and respond to us. We appreciate that you still feel positive about our paper. We are wondering if our rebuttal has resolved all of your concerns. We are happy to clarify more if you have any remaining concerns. We are also wondering if you would like to consider raising your score in light of the new experiments showing the real-world use case of the proposed algorithm and the clarification in the rebuttal materials.

---

### Official Review · Reviewer_8JqR · 2023-07-03

**Soundness:** 3 good
**Presentation:** 3 good
**Contribution:** 3 good
**Rating:** 6
**Confidence:** 4

**Summary:**

The paper introduces a novel demonstration selection method aimed to enhance performance of (few-shot) in-context learning. The approach is characterized by a two-stage process that includes latent concept learning and demonstration selection, each holding a unique significance. In the latent concept learning stage, the authors present a method for acquiring a task-specific token embedding set through prompt-tuning. Subsequently, the demonstration selection stage involves the process of selecting in-context samples. This process is based on maximizing the likelihood of post-fixing the previously acquired task latent. The efficacy of the demonstration selection has been evaluated across various language models, leading to improved in-context learning performance.

**Strengths:**

This paper is clearly written with a clear definition of the task.
The proposed method is presented with great clarity and detail, which significantly aids in understanding the overall procedure.
The paper presented impressive performance results. The successful transfer of demonstrations selected from a smaller model (GPT-2) to other larger models is very impressive.
There are extensive ablation and additional experiments for deeper analysis of each component of the proposed method.

**Weaknesses:**

An area of concern lies in the assumption that the task latent derived through prompt-tuning is considered the “optimal task latent” (line 146). This assumption may not hold universally, and could be re-considered as the use of manual prompts - which could offer a more intuitive understanding of the task.

**Questions:**

- Considering that the order of samples in the demonstration does not play a critical role in the proposed method, what motivated the decision to include a "re-ordering" (selecting the permutation) step in the demonstration selection phase?
- How would the performance change if a larger model, as opposed to GPT-2, was employed for demonstration selection? Would this potentially enhance the overall performance?


**Limitations:**

Limitations are stated in the appendix C.

---

> ### Author Rebuttal · Authors · 2023-08-09
>
> Thank you for your positive review. Below is our response to your comments:
>
> 1. **About optimal latent**: Yes, in practice, the learned latent is not (even never) optimal, but approximate. The assumed optimality is for deriving the upper bound theory that the in-context learning classifier can be as good as the Bayes optimal classifier. And the manual prompt can also be viewed as an approximate of the optimal latent. However, by using a manual prompt, we are not able to prove our theory of in-context learning that LLMs infer a latent variable at the inference time. The prompt tuning approach allows us to reveal the unobservable latent task variable in LLMs.
>
> 2. **About reordering**: The reordering step in the proposed algorithm is indeed redundant. It’s a leftover from the original algorithm design before we perform the ablation study. We will remove this step in the paper.
>
> 3. **About better model for demonstration selection**: In point 2 of our general response, we show that a better/larger model can select better demonstrations with the GSM8K dataset. More specifically, our method with Llama 2 (7B) outperforms our method with GPT2-XL (1.3B) by 0.8-3.4% absolute accuracy when performing in-context learning with different models.

---

> > ### Comment · Reviewer_8JqR · 2023-08-19
> >
> > I appreciate your insightful response. The authors' clarification regarding the concept of "optimal latent" and the inclusion of additional experiments have certainly contributed to my understanding of the presented paper. Still, I find that the theoretical assumptions and the detailed analysis of the conducted experiments, which support the reported enhancements in ICL performance, could benefit from further elaboration. In light of this, my assessment remains aligned with the initial score assigned.

---

### Official Review · Reviewer_sfNW · 2023-07-06

**Soundness:** 2 fair
**Presentation:** 2 fair
**Contribution:** 3 good
**Rating:** 5
**Confidence:** 3

**Summary:**

This paper describes a framework to select demonstrations for in-context learning by using bayseian formulation for the data generation process. Similar to topic models, the formulation uses a "concept" variable and words in this sequence are conditioned over this concept variable and are conditionally independent of the other tokens in the generated text. The concept variable models the prompt/task instruction and is modeled by learning concept tokens by prompt tuning a small LLM. Since the concept tokens are made part of the vocabulary, the selection of in-context demonstrations can be learnt by maximizing the probability for the concept tokens. Experiments are presented on multiple NLP tasks and they indicate that selecting in-context demonstrations using this formulation works better than using random in-context examples.

**Strengths:**

1. Simple formulation that aids better selection of in-context demonstrations
2. Experiments have been presented on multiple NLP tasks

**Weaknesses:**

Modeling: I'm not sure the topic-model like bag-of-words assumption is accurate in the way the modeling has been described. As Line 56 in the paper states, the generation of tokens would be independent of the previous tokens but to truly model this wouldn't you need to modify the current latent concept learning setup to be sequence-agnostic? The method works empirically so perhaps its okay but some clarification would help here. Perhaps the authors can explain further in the rebuttal phase if I misunderstood it.

Experiments: The experiment baselines refer to the "Similar" baseline but any discussion or analysis of its results is completely missing from the main paper. I also checked the appendix, and the results in Table 3 (appendix) do not correspond to the results reported in the main paper in the histogram plots.

Notation: The notation is a bit hard to follow and could benefit from simplification.

**Questions:**

See weakness

**Limitations:**

Yes

---

> ### Author Rebuttal · Authors · 2023-08-09
>
> Thank you for your review. Below is our response to your comments:
>
> 1. **About topic model assumption**: We want to clarify that the topic model here is in a more general sense, which is not equivalent to LDA [1], but similar to the modern neural topic models proposed in [2,3]. In this more general definition of the topic model, the tokens/words are not required to be conditionally independent given the topic variable. We are aware that this definition of a topic model is basically a simple latent variable model of language with a single latent. Thanks for pointing this out, to avoid further confusion, we decided to change our title to Large Language Models as Latent Variable Models: Explaining and Finding Good Demonstrations for In-Context Learning. We will also revise our paper to make this point clearer.
>
> 2. **About Similar baseline**: We define the "Similar" baseline at line 238,  which means using the most similar examples to the current query as the demonstrations for in-context learning, and it was first proposed in [4]. Since it is relatively straightforward and is a default demonstration selection method that has been used in many papers, we did not include much analysis of it in the main paper. Thanks for pointing this out. In general, our experiment results show that using demonstrations similar to the testing query will improve the in-context performance compared to random selection, which implies that similar examples contain useful information for LLMs to infer about some parts of the task latent that is relevant to the current query. However, it is still not as effective as directly selecting the examples that can best infer the whole task latent, as used in our proposed method, which can cover more aspects of the task. We will add this analysis to the revision.
>
> 3. **About the main results**: Figure 2 in the main paper corresponds to the last column of Table 3 in the appendix. You probably looked at the last three rows (which correspond to the average over all the models) instead of the last column (which corresponds to the average over all the datasets). We will revise the table to make it clearer.
>
> [1] David M. Blei, Andrew Y. Ng, and Michael I. Jordan. Latent dirichlet allocation. J. Mach. Learn. Res. 2003.
>
> [2] Miao, Yishu, Edward Grefenstette, and Phil Blunsom. Discovering discrete latent topics with neural variational inference. ICML 2017.
>
> [3] Miao, Y., Yu, L. & Blunsom, P. Neural Variational Inference for Text Processing. ICML 2016.
>
> [4] J. Liu, D. Shen, Y. Zhang, B. Dolan, L. Carin, and W. Chen. What makes good in-context examples for GPT-3? In Proceedings of Deep Learning Inside Out (DeeLIO 2022): The 3rd Workshop on Knowledge Extraction and Integration for Deep Learning Architectures

---

> > ### Comment · Reviewer_sfNW · 2023-08-18
> > **Acknowledgement of Rebuttal**
> >
> > Thank you for your response. I re-read my review in light of your responses. I had understood what the similar baseline refers to but I had missed its inclusion in Figure 2. I'm updating my review score

---

> > > ### Author Response · Authors · 2023-08-18
> > >
> > > Thank you for reading our rebuttal and responding. We are glad that our rebuttal helped clarify your concerns. And we appreciate your raise of the score. Have a good weekend :)!

---

### Official Review · Reviewer_XxTq · 2023-07-17

**Soundness:** 3 good
**Presentation:** 3 good
**Contribution:** 3 good
**Rating:** 5
**Confidence:** 3

**Summary:**

This paper tries to study in-context learning through a Bayesian lens, namely treating LLMs as implicit topic models. It proposes an algorithm to select optimal demonstrations from a set of annotated data with a small LLM and then use the selected demonstrations with larger LLMs. 12.5% improvement compared to random selection is observed by adopting the proposed algorithm.

**Strengths:**

The paper has extensive empirical results showing the proposed algorithm can effectively find helpful demonstration examples to boost the performance of in-context learning.

**Weaknesses:**

Although we see significant performance improvement, it's hard to conclude LLMs are topic models. It could be possible that topic (or distribution of input text) is one of the  important factors but there are other attributes affecting in-context learning. In some ways, this also conflicts with some previous work, e.g. "In-context Learning and Induction Heads" points out the pattern copying behavior is the key, "Robustness of Demonstration-based Learning Under Limited Data Scenario" finds out random tokens are also helpful.

**Questions:**

Have you tried to compared with better algorithms to find demonstrations, e.g. "Selective Annotation Makes Language Models Better Few-Shot Learners"?

**Limitations:**

yes

---

> ### Author Rebuttal · Authors · 2023-08-09
>
> Thank you for your review. Below is our response to your comments:
>
> 1. **About our claims**: We would like to clarify that we are not claiming that LLMs are topic models. Our claim is that LLMs are IMPLICITLY topic models, that learn and infer a latent task/topic variable for solving each query. We would be very happy to revise and clarify this.
>
> There are many plausible ways of understanding in-context learning. The topic model/latent variable view is indeed just one important piece of it. We in no way claim ours is the only correct one. We would like to clarify that many of these are complimentary views of LLMs that do not necessarily conflict with either other. For example, the in-context learning as gradient descent view [6,7,8] can be understood as a specific way of inferring and utilizing the task latent at in-context learning time, as in the Bayesian interpretation framework [9,10,11].
>
> 2. **About conflicting findings**: The pattern information of a task can be viewed as also included in the inferred latent variable, so our explanation is not conflicting [1]. There is a similar finding in [3] as the mentioned random tokens finding in [2], saying that the demonstrations with random labels also work. In contrast, [4] explicitly argues that ground truth labels matter, while the sensitivity to label correctness varies across different tasks. Interestingly, [3] and [4] are published at the same conference. The experiment in Table 7 of our appendix shows that the correct label does matter on our tasks, with or without our proposed demonstration selection method, which agrees with [4]. As the results in [2] are mostly from the NER task, it is likely that NER has a high tolerance for randomness in the demonstrations.
>
> 3. **About baselines**: We are aware of the mentioned paper [5] and have cited it in our related work section. We did not include it as a baseline because its setting is too different from ours, which makes it incomparable to our method. They propose a two-phase algorithm, where the first phase selectively annotates a relatively small number (18 to 100) of data from a large set of unlabeled data (3K), and then the second phase retrieves demonstrations from the annotated data based on their similarity to the query. We are, on the other hand, exclusively focusing on selecting demonstrations from a small set of labeled data (100). So our proposed method is only comparable with the second phase of their algorithm, which is the same as the ‘Similar’ baseline (defined in line 238) used in our experiments. It is possible to combine our method with the first phase of [5] to obtain a better performance under the selective data annotation setting with a large set of unannotated data, but this setting is out of the scope of our current paper, so we leave this to the future work.
>
> [1] Olsson, Catherine, et al. In-context learning and induction heads. arXiv preprint arXiv:2209.11895 (2022).
>
> [2] Hongxin Zhang, Yanzhe Zhang, Ruiyi Zhang and Diyi Yang. Robustness of Demonstration-based Learning Under Limited Data Scenario. EMNLP 2022
>
> [3] Min, Sewon, et al. Rethinking the Role of Demonstrations: What Makes In-Context Learning Work?. EMNLP 2022.
>
> [4] Yoo, Kang Min, et al. Ground-Truth Labels Matter: A Deeper Look into Input-Label Demonstrations. EMNLP 2022.
>
> [5] Su, Hongjin, et al. Selective annotation makes language models better few-shot learners. ICLR 2022.
>
> [6] E. Akyürek, D. Schuurmans, J. Andreas, T. Ma, and D. Zhou. What learning algorithm is in-context learning? investigations with linear models. NeurIPS 2022.
>
> [7] D. Dai, Y. Sun, L. Dong, Y. Hao, Z. Sui, and F. Wei. Why can gpt learn in-context? language models secretly perform gradient descent as meta optimizers. ACL 2023.
>
> [8] J. von Oswald, E. Niklasson, E. Randazzo, J. Sacramento, A. Mordvintsev, A. Zhmoginov, and M. Vladymyrov. Transformers learn in-context by gradient descent. ICML 2023.
>
> [9] M. Hahn and N. Goyal. A theory of emergent in-context learning as implicit structure induction. arXiv preprint, 2023.
>
> [10] H. Jiang. A latent space theory for emergent abilities in large language models. arXiv preprint 2023.
>
> [11] S. M. Xie, A. Raghunathan, P. Liang, and T. Ma. An explanation of in-context learning as implicit bayesian inference. ICLR 2022.

---

> > ### Author Response · Authors · 2023-08-20
> >
> > As the end of discussion period is approaching, we are wondering if you have read our rebuttal and if you have any remaining concerns. We are happy to clarify more before the discussion period ends.

---

### Official Review · Reviewer_waNU · 2023-07-26

**Soundness:** 3 good
**Presentation:** 3 good
**Contribution:** 3 good
**Rating:** 5
**Confidence:** 5

**Summary:**

This work aims at proposing a demonstration example selection algorithm in in-context learning, using a formulation of topic models for language models. Specifically, the proposed algorithm applies prompt tuning on some prefixing learnable tokens to obtain latent concepts, and select demo examples that maximize the probability of inferring the learned tokens. The authors test the algorithm on small LMs and find generalizability to larger LMs, on a set of classification tasks.

**Strengths:**

The writing of the paper is overall clear. The theoretic formulation is well-structured and points out where assumptions and approximations are made. The proposed algorithm is tested on a set of tasks and models. The ablation studies are also reasonable.

**Weaknesses:**

The motivation and takeaway of this work are rather vague. As a demonstration example selection algorithm, the setup is not realistic enough since it uses prompt tuning with labeled task data to obtain the latent concept tokens. Subsequently, the comparison with the baseline methods using uniform or similar demonstration examples is not fair, since they do not assume prompt tuning with labeled task data.

Some natural and probably necessary questions here include: (1) how does the performance of the proposed method compare with the vanilla prompt tuning performance? and (2) since labeled task data are used, what is the performance of directly (and independently) select ICL examples based on their contribution to the predictions of the labeled task data?

Additionally, the current analysis is mostly on the performance of the algorithm rather than the selected demonstration examples themselves. What attributes do they share in common? (apart from the qualitative clustering shown in the tSNE figure) What label distribution do they have? (do they simply act as a *calibration* to the model's output distribution?) In other words, probably as the title hints, what are good demonstrations for in-context learning?

**Questions:**

Please see the weakness section for my main questions to the authors. Additionally, is any generation task considered in this work apart from the classification tasks?

---

> ### Author Rebuttal · Authors · 2023-08-09
>
> Thank you for your detailed review. Below is our response to your comments:
>
> 1. **Takeaway of our paper**: We would like to clarify that our study's focus is on elucidating the underlying mechanisms of in-context learning and presenting a novel approach to demonstration selection that empirically verifies our theory in a real-world setting. By employing prompt tuning with labeled data, we can tease apart latent variables and relationships that otherwise might be obscured.
>
> 2. **Real-world use case and generation task**: In short, our proposed method is most useful with hard tasks that smaller LLMs perform significantly worse than larger models. Please refer to point 2 of our general response for detailed results on GSM8K. In this case, even parameter-efficient fine-tuning with smaller models cannot obtain reasonable performance (less than 4% accuracy), while in-context learning combined with our method can obtain significantly higher performance with both small and large models. Our method can obtain more than 80% accuracy with ChatGPT, compared with 76.5% accuracy using random selection. Note that our method only requires tuning a small model on limited data, so it is data and computation efficient. We want to reiterate that improving and connecting a previously synthetic-only theory of in-context learning [6] with real-world models and data is already non-trivial and is relatively rare in the current theories of LLMs, which are usually disconnected from the real world. The value of our paper does not only lie in its real-world application.
>
> 3. **Baselines**: To the best of our knowledge, none of the existing demonstration selection methods involves tuning a smaller model. So we are not able to involve a similar baseline. About the proposed contribution-based baseline, we are not quite sure what the reviewer means by ‘select ICL examples based on their contribution to the predictions of the labeled task data’. Do you mean select demonstrations by attributing the prompt-tuned model to the labeled training data? If that is the case, we agree this is an interesting idea, which to the best of our knowledge, has not been used on any existing demonstration selection method yet. However, we want to argue that this baseline is too exploratory, which could be a new project by itself, and does not seem to be natural enough to be incorporated in the current paper. We think this is out of the scope of our project, but we are happy to discuss it with the reviewer if they can clarify what they mean here.
>
> 4. **Compared to prompt tuning**: For simpler tasks like sentiment analysis, prompt tuning performs better than in-context learning, even when combined with our method (detailed results see Figure 8 in the Appendix). For harder tasks like GSM8K, simple prompt tuning with small models cannot obtain meaningful results (less than 4% accuracy), while in-context learning can obtain significantly higher performance with both large models (more than 80% accuracy with ChatGPT combined with our method) and small models (19.3% accuracy with Llama2 7B combined with our method). For detailed results, please refer to point 2 of our general response.
>
> 5. **Analysis of the selected demonstrations**: We didn’t include an analysis of the selected demonstration examples in the paper because the common features shared between the selected examples are a bit hard to detect. However, we agree that it is still necessary to include such an analysis. Because of the space limitation, we only list the top demonstrations of GSM8K and SST2 in the supplemental 1-page pdf.
>
> Compared to the examples with lower scores, the selected examples for GSM8K have more deductive reasoning (i.e. with the connecting words ‘so’, ‘then’, ‘thus’, etc.), instead of listing parallel conditions. For SST2, the selected examples are longer and more complex, sometimes including a ‘but’. This can be understood as these harder examples can represent the task more comprehensively. This conclusion also aligns with the finds in [1] that hard examples in the pre-training data contribute to in-context learning the most. The label distribution of the selected demonstrations is usually balanced in class, which reduces the possible biases introduced by the demonstrations. We will add the analysis of all selected demonstrations in the revision.
>
> [1] Han, Xiaochuang, et al. Understanding In-Context Learning via Supportive Pretraining Data. ACL 2023.

---

> > ### Author Response · Authors · 2023-08-19
> >
> > As the end of the discussion period is approaching, we just want to make sure that you have read our rebuttal. It would be great if you can clarify some of your comments that we didn't understand (i.e. *‘select ICL examples based on their contribution to the predictions of the labeled task data’*) so that we can give proper responses. We are also more than happy to clarify if there are any remaining concerns.

---

> ### Comment · Reviewer_waNU · 2023-08-19
> **Thanks for the response**
>
> Thanks for your detailed response. Below I'll first clarify my concerns over the selection of the baselines (Re: 3, 4) and then the overall takeaway of the work (Re: 1, 5).
>
> The authors described their method clearly in Figure 1, with two main stages: (a) Use prompt tuning to obtain concept tokens. During this process, a *labeled* dataset D is used (Algorithm 1). (b) Select k demonstration data from a candidate data set D^d (Algorithm 2). The size of D^d is 100 as mentioned in Line 230. However, it is not immediately clear to me what the size of D is, though the authors mentioned it is "limited data" (the method "requires tuning a small model on limited data"; can you perhaps clarify further?). My concern is that the two baselines compared in this work, random selection and selection based on similarity, did not utilize this labeled dataset D. Therefore in my review, I suggested two more comparisons: (1) Compare with prompt tuning as it uses the same labeled dataset D. (2) Since prompt tuning does not involve D^d in an ICL setup, I mentioned "since labeled task data are used, what is the performance of directly (and independently) select ICL examples based on their contribution to the predictions of the labeled task data". To clarify this a bit further, I meant to select/prepend demonstration examples from D^d based on whether they can maximize the probability of generating labeled examples from D (i.e., using the terminology from Algorithm 1 and 2 --- P(Y | X^d, Y^d, X) ). Having these two comparisons that also utilize the labeled dataset D will help the audience understand the importance of deriving the concept tokens theta more clearly.
>
> My second concern, apart from the demonstration selection algorithm, is on the takeaway of this work. The authors clarified that the goal is "elucidating the underlying mechanisms of in-context learning". In that way, I think a deeper analysis into the selected demonstration examples for each task would be necessary, to give the audience an interpretable picture of the mechanism of ICL.
>
> I agree that the transferability of the demonstration examples from small to large models is interesting and potentially useful (Re: 2). I have changed my scores accordingly.

---

> > ### Author Response · Authors · 2023-08-20
> >
> > Thank you for carefully reading through our rebuttal and responding to us. We appreciate your raise of the score. Below is our response to your concerns:
> >
> > 1. **About the size of D**: The size of D range from 346 (ETHOS-SO, ETHOS-R) to 1.6k (SST2, FPB, COLA, DBpedia, EmoC, EmoS, GSM8K), which is determined by the availability of the annotated data and then caped by 1.6k. We will state this explicitly in the revision.
> >
> > 2. **About prompt tuning baseline**: We involved the comparison with prompt tuning on D in Figure 8 in the Appendix and also in the new experiments with GSM8K, as detailed in point 4 in our rebuttal. We will add this in the main paper in the revision.
> >
> > 3. **About contribution-based baseline**: We now understand the second baseline proposed by the reviewer. Thank you for the clarification. We will run the suggested baseline and either post the results here if we can get it done before the discussion period ends on August 21 or we will directly add this baseline in the revision.
> >
> > 4. **About in-depth analysis of the selected demonstrations**: We will involve an analysis of the selected demonstration from all datasets, both from a qualitative perspective as shown in the last paragraph of our rebuttal, and from a qualitative perspective by analyzing the text distribution and information gain of the selected demonstrations, similar to [1]. We will either post the quantitative results here if we can get it done before the discussion period ends on August 21 or we will directly add this in the revision.
> >
> > [1] Han, Xiaochuang, et al. Understanding In-Context Learning via Supportive Pretraining Data. ACL 2023.

---

### Author Rebuttal · Authors · 2023-08-09

We want to first thank all the reviewers for taking the time to review our paper. We provide some clarification and additional results below:

1. **About topic modeling assumptions**: We want to clarify that the topic model here is in a more general sense, which is not equivalent to LDA [1], but similar to the modern neural topic models proposed in [2,3]. In this definition, the tokens/words are not required to be conditionally independent given the topic variable. We know that this topic model definition is essentially a simple latent variable model of language with a single latent. To avoid further confusion, we will change our title to **Large Language Models Can be Viewed as Latent Variable Models: Explaining and Finding Good Demonstrations for In-Context Learning**. We do not intend to claim that this latent variable explanation of in-context learning is the only correct one. As described in the related work section, multiple plausible and complementary ways exist to understand and interpret in-context learning.

2. **About the realisticness of our setting**: Since our primary goal is to connect the theory with real-world models and datasets, we did not try to include harder tasks. In practice, our proposed method is most effective with hard tasks that even parameter-efficient fine-tuning with smaller models cannot outperform in-context learning with the same or larger models. To improve the performance under a low data setting, with a small computing budget, and minimal inference latency, our demonstration selection method is a reasonable choice. Our demonstration selection method can also potentially be combined with other prompting techniques to boost performance further.

**We added a new dataset, GSM8K** [4], which is a math word problem-solving dataset with chain-of-thoughts solutions. The table below shows the test accuracy of the final numerical answer using greedy generation. Note that we did not use a calculator to insert the correct result of each generated math equation during generation for time efficiency, which resulted in slightly lower scores.

As shown in the first row of the table, prompt tuning with ten new tokens can only obtain less than 4% accuracy on the GSM8K test set. While it is possible that a better-designed efficient tuning algorithm with more trainable parameters will be able to get better results, it is still unlikely that the efficient tuning results with small models will be able to outperform the few-shot performance of larger/better models,  as [8] show that a fully fine-tuned Llama 7B model on the whole GSM8K training set can only get 35.9% accuracy on GSM8K, and can be improved to 49.3% if combined with data augmentation, which is still significantly lower than the over 80% accuracy obtained by ChatGPT combined with our method.

The last 4 rows show the in-context learning results with different size Llama 2 models [5] and ChatGPT. Our proposed demonstration selection method (last two columns) significantly outperformed the Uniform and Similar baseline as defined in line 236 to line 247 in our paper. Also, note that the demonstrations selected with a larger model (7B) are more effective than those selected with a smaller model (1.5B).

|  | Uniform | Similar | Ours w/ Llama 2 (7B) | Ours w/ GPT2-XL (1.5B) |
| --- | --- | --- | --- | --- |
| Prompt tuning | N/A | N/A | 3.7 | 1.3 |
| Llama 2 (7B) / 4-shot | 11.4 | 13.1 | **19.3** | 15.9 |
| Llama 2 (13B) / 4-shot | 17 | 18.3 | **21.6** | 20.5 |
| Llama 2 (70B) / 4-shot | 50.2 | 53.5 | **54.3** | 52.9 |
| ChatGPT (gpt-3.5-turbo) / 4-shot | 76.5 | 78.1 | **81.2** | 80.4 |

*We want to reiterate that improving and connecting a previously synthetic-only theory of in-context learning [6] with real-world models and data is already non-trivial and is relatively rare in the current theories of LLMs, which are usually disconnected from the real world. The value of our paper does not only lie in its real-world application.*

3. **Analysis of the selected demonstrations**: We didn’t include an analysis of the selected demonstration examples in the paper because the common features shared between the selected examples are a bit hard to detect. However, we agree that it is still necessary to include such an analysis. Because of the space limitation, we only list the top demonstrations of GSM8K and SST2 in the supplemental 1-page pdf.

Compared to the examples with lower scores, the selected examples for GSM8K have more deductive reasoning (i.e. with the connecting words ‘so’, ‘then’, ‘thus’, etc.), instead of listing parallel conditions. For SST2, the selected examples are longer and more complex, sometimes including a ‘but’. This can be understood as these harder examples can represent the task more comprehensively. This conclusion also aligns with the finds in [7] that hard examples in the pre-training data contribute to in-context learning the most. The label distribution of the selected demonstrations is usually balanced in class, which reduces the possible biases introduced by the demonstrations.

[1] David M. Blei, Andrew Y. Ng, and Michael I. Jordan. Latent dirichlet allocation. J. Mach. Learn. Res. 2003.

[2] Miao, Yishu, Edward Grefenstette, and Phil Blunsom. Discovering discrete latent topics with neural variational inference. ICML 2017.

[3] Miao, Y., Yu, L. & Blunsom, P. Neural Variational Inference for Text Processing. ICML 2016.

[4] Cobbe, Karl, et al. Training verifiers to solve math word problems. arXiv preprint 2021.

[5] Touvron, Hugo, et al. Llama 2: Open foundation and fine-tuned chat models. arXiv preprint 2023.

[6] Xie, Sang Michael, et al. An Explanation of In-context Learning as Implicit Bayesian Inference. ICLR 2021.

[7] Han, Xiaochuang, et al. Understanding In-Context Learning via Supportive Pretraining Data. ACL 2023.

[8] Yuan, Zheng, et al. Scaling Relationship on Learning Mathematical Reasoning with Large Language Models. arXiv preprint 2023.

---

### Author Response · Authors · 2023-08-15

Dear Reviewers,

Thank you for your valuable review. We have provided responses to your questions, and are committed to address further concerns.

We would like to ask for your kind participation in the discussions. Please let us know if we have addressed your concerns or if you have additional feedback or suggestions.

We highly appreciate your time and efforts and are looking forward to the discussions.


Best regards,
Authors

---

### Decision · Program_Chairs · 2023-09-21

**Decision:**

Accept (poster)

**Comment:**

This paper explores the in-context learning capability of large language models (LLMs) through a Bayesian lens, treating LLMs as implicit topic models. The authors propose a two-stage algorithm that first learns the latent concepts using prompt tuning and then selects optimal demonstrations from labeled data with a smaller language model. The method yields improvements compared to random selection, across several models and datasets.

The reviewers are generally positive about this work. They praise that the paper is clear and well-written, the proposed method shows improvements over baselines, and experiments performed on multiple datasets.
However, there are some concerns regarding the manuscript:
Some reviewers felt the paper lacks a complete analysis, either of the algorithm's performance (waNU) or of the experiment baselines (sfNW). XxTq, sfNW, and 8JqR raise concerns about the methodological and theoretical assumptions made in the paper.
XxTq pointed out conflicts with previous research. sfNW questioned the bag-of-words assumption. 8JqR expressed concerns over the "optimal task latent" assumption.Reviewer uvg2 raised multiple issues including the problem setting being unrealistic due to the amount of training data assumed, and the simplicity of the evaluation tasks. While some of these issues are addressed in rebuttal, it seems like most reviewers remained only slightly positive about the work.